# DiSRouter: Distributed Self-Routing for LLM Selections

**Hang Zheng**[1][*] **Hongshen Xu**[1][*]**, Yongkai Lin**[3,4]**, Shuai Fan**[3,4]**, Lu Chen**[1,2,3,5][†] **Kai Yu**[1,3,5][†]

[1]X-LANCE Lab, School of Computer Science, Shanghai Jiao Tong University, Shanghai, China
[2]Shanghai Innovation Institution, Shanghai, China
[3]Jiangsu Key Lab of Language Computing, Suzhou, China
[4]AISpeech Co., Ltd., Suzhou, China
[5]Suzhou Laboratory, Suzhou, China
`{azure123, chenlusz, kai.yu}@sjtu.edu.cn`

## Abstract

The proliferation of Large Language Models (LLMs) has created a diverse ecosystem of models with highly varying performance and costs, necessitating effective query routing to balance performance and expense. Current routing systems often rely on a centralized external router trained on a fixed set of LLMs, making them inflexible and prone to poor performance since the small router can not fully understand the knowledge boundaries of different LLMs. We introduce *DiSRouter* (Distributed Self-Router), a novel paradigm that shifts from centralized control to distributed routing. In *DiSRouter*, a query traverses a network of LLM agents, each independently deciding whether to answer or route to other agents based on its own self-awareness—its ability to judge its competence. This distributed design offers superior flexibility, scalability, and generalizability. To enable this, we propose a two-stage Self-Awareness Training pipeline that enhances each LLM's self-awareness. Extensive experiments demonstrate that *DiSRouter* significantly outperforms existing routing methods in utility across various scenarios, effectively distinguishes between easy and hard queries, and shows strong generalization to out-of-domain tasks. Our work validates that leveraging an LLM's intrinsic self-awareness is more effective than external assessment, paving the way for more modular and efficient multi-agent systems.

## 1 Introduction

The proliferation of Large Language Models (LLMs) has marked a new era, demonstrating remarkable capabilities across tasks. However, the superior performance of state-of-the-art LLMs is often accompanied by prohibitive computational and financial costs, limiting their widespread adoption. Concurrently, a rich and diverse ecosystem of LLMs has emerged, ranging from small, efficient models for edge deployment to powerful, cloud-based LLMs (Chen et al., 2025). This diversity presents both a critical challenge and a significant opportunity: how to navigate the LLMs pool to select the most cost-effective agent for a given query without compromising performance?

This challenge, often termed "query routing" or "model selection," has drawn significant research attention (Varangot-Reille et al., 2025) and found industrial application, with systems like the latest GPT-5 employing a unified architecture with multiple models and a real-time router (OpenAI, 2025). Conventional approaches typically utilize a centralized router, such as a scoring model, to assess query difficulty or predict the potential performance of a specific LLM, subsequently dispatching queries to the smallest capable model (Ong et al., 2024; Chen et al., 2023; Feng et al., 2024). Despite their effectiveness, these centralized systems suffer from two fundamental limitations: (1) **Limited Flexibility of External Routers.** External routers are usually trained on a fixed set of candidate LLMs. Once trained, any modification, such as adding or updating an agent, necessitates a costly retraining of the entire routing system, which makes the system rigid and undermines scalability.

---

[*]Hang Zheng and Hongshen Xu contribute equally to this work.
[†]Lu Chen and Kai Yu are the corresponding authors.

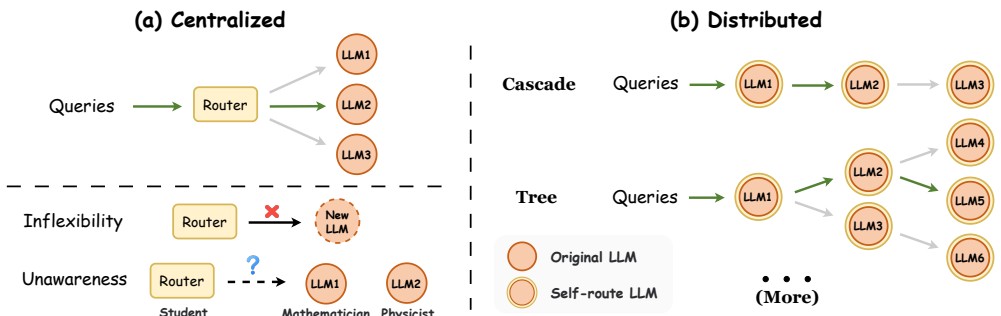

Figure 1: **Comparison between (a) centralized and (b) distributed routing.** Centralized routing relies on an external router, which has limitations in adapting to new LLMs (Inflexibility) and accurately assessing the knowledge boundaries of each LLM (Unawareness). In contrast, distributed routing is not dependent on an external router and can be organized into various structures.

(2) **Inaccurate Capability Assessment.** External routers are often implemented as relatively small models. Lacking the capacity to fully comprehend the intrinsic knowledge boundaries of large-scale LLMs, they struggle to accurately assess the capabilities of each model for a given query and determine the most suitable model. Consequently, their limited capability of assessment often becomes a major bottleneck for the routing system.

To overcome these limitations, we propose a paradigm shift from centralized control to distributed routing, as shown in Figure 1. We introduce *DiSRouter* (Distributed Self-Router), a novel framework that replaces a single central router with a network of agents capable of self-assessment and autonomous routing decisions. The core idea is to empower each agent to evaluate its own competence on a given query and independently decide whether to answer it or delegate it onward, rather than relying on an external router to make routing decisions. In our experiments, we instantiate *DiSRouter* as a cascade of homogeneous LLM agents of increasing sizes. Each smaller agent first attempts to assess and solve the query; if it determines that it cannot provide a reliable answer, it autonomously routes the query to the next larger agent in the cascade. This design fosters a fully distributed, flexible, and modular system that naturally supports the seamless addition or removal of agents. While our evaluation focuses on the cascade architecture, the *DiSRouter* framework is not limited to this setting. In principle, *DiSRouter* can be instantiated with alternative network topologies, such as tree or mesh structures, a prospect we explore further in the discussion section §6.

The effectiveness of *DiSRouter* highly depends on the self-awareness of each agent, namely its ability to accurately assess the boundaries of its own capabilities. To this end, we propose a Self-Awareness Training pipeline comprising Supervised Fine-Tuning (SFT) followed by Reinforcement Learning (RL). A key innovation is the design of a localized reward function that enables each agent to be trained completely independently and in parallel. To further align the model with diverse scenario requirements, the reward function incorporates a global preference factor ($\alpha$), allowing the entire system to adaptively shift its collective behavior from a performance-first to a cost-first stance based on user preference, without requiring any direct inter-agent communication.

Our contributions are summarized as follows:

- We propose *DiSRouter*, a novel distributed, self-routing system that achieves superior modularity, scalability, and robustness compared to traditional centralized routers.
- We design a Self-Awareness Training pipeline with a localized reward that enables fully independent and parallel training of reliable agents in a distributed routing system.
- Our extensive experiments demonstrate that *DiSRouter* effectively balances performance and cost, while dynamically adapting its routing strategies to user-defined preferences.

## 2 PROBLEM FORMULATION

The primary objective of a routing system is to develop an optimal strategy that assigns each incoming query to the most suitable agent, effectively balancing system performance and cost.

## 2.1 Routing System

A routing system comprises three main components: a set of queries $X$, a pool of models $M$, and a routing policy $\pi$. The policy $\pi$ maps each query $x_j$ to a specific agent $\pi(x_j) \in M$, denoted as $\pi : X \to M$. Let's consider a set of $N$ queries $X = \{x_i\}_{i=1}^N$ and a pool of $K$ LLMs $M = \{m_i\}_{i=1}^K$, where $X$ is drawn from a data distribution $D$. We define the score of model $m_i$ on query $x_j$ as $A(m_i, x_j) \in \{0, 1\}$, where 1 denotes a correct solution and 0 otherwise. To facilitate a comparison of costs from different model sources, such as API call prices or GPU seconds, we assign a predefined and fixed cost $c_i \in [0, 1]$ to each model $m_i$. Without loss of generality, we assume the models are ordered by increasing cost, such that $m_1$ has the lowest cost and $m_K$ has the highest cost. In addition to the inference cost of the models, the routing process itself incurs an extra overhead, which is typically negligible (as discussed in Appendix A.1), and is therefore not considered in this work. Thus, the total cost function for an agent $m_i$ is simply $C(m_i) = c_i$.

The system's overall performance and cost can then be calculated as follows, where $\hat{m}_i = \pi(x_i)$:

$$Performance = \frac{1}{N} \sum_{i=0}^N A(\hat{m}_i, x_i) \qquad Cost = \frac{1}{N} \sum_{i=0}^N C(\hat{m}_i) \qquad (1)$$

We adopt a widely used metric, **Utility**, to quantify the effectiveness of a routing policy by combining both Performance and Cost (Tsiourvas et al., 2025; Xu et al., 2025), which is defined as:

$$Utility = Performance - \alpha \cdot Cost \qquad (2)$$

where $\alpha \in [0, 1]$ is a hyperparameter named "preference factor" representing the user's preference between performance and cost. A smaller $\alpha$ prioritizes accuracy, while a larger $\alpha$ emphasizes cost efficiency. Specifically, we define three scenarios referencing that in *GraphRouter* (Feng et al., 2024): "Performance First", "Balance", and "Cost First", with $\alpha$ set to 0.2, 0.5, and 0.8, respectively.

## 2.2 Routing System Architecture

In this section, we delineate the architectures of centralized and our proposed distributed routing.

### 2.2.1 Centralized Routing

In a centralized routing architecture, a single entity, typically a trained model or a set of predefined rules, acts as the central router. This router embodies the routing policy $\pi$ and independently assigns each query to a specific model in the pool. In this setup, the overarching goal of the router is to learn an optimal routing mapping $\pi^*$ that maximizes the expected utility over the data distribution $D$:

$$\pi^* = arg\max_\pi \; E_{x \sim D}[A(\pi(x), x) - \alpha \cdot C(\pi(x))] \qquad (3)$$

### 2.2.2 Distributed Routing

In a distributed architecture, the system operates without a central router. Instead, the pool of agents forms a routing network where each agent independently decides whether to execute the query itself or delegate it to another agent. In this setup, the routing policy $\pi$ is not a single, monolithic function. Instead, it is comprised of local policies distributed among each agent, denoted as $\Pi = \{\pi_1, \pi_2, \ldots, \pi_K\}$, where each $\pi_i$ is the local policy for agent $m_i$. For a given query $x$, the policy $\pi_i(x)$ selects a target agent $m_j$ from a set of possible actions. If the agent selects itself ($j = i$), it commits to executing the query. If it selects another agent ($j \neq i$), it routes the query to $m_j$. To simplify the evaluation, we enforce that an agent can only route to agents of a higher index (and thus higher cost). The action space for agent $m_i$ is therefore:

$$\pi_i(x) \in \{m_j \in M \mid j \geq i\} \qquad (4)$$

To guarantee that every query is eventually processed, the policy of the final agent, $m_K$, is fixed to always execute, i.e., $\pi_K(x) = m_K$ for all $x$, serving as a fallback or "expert of last resort."

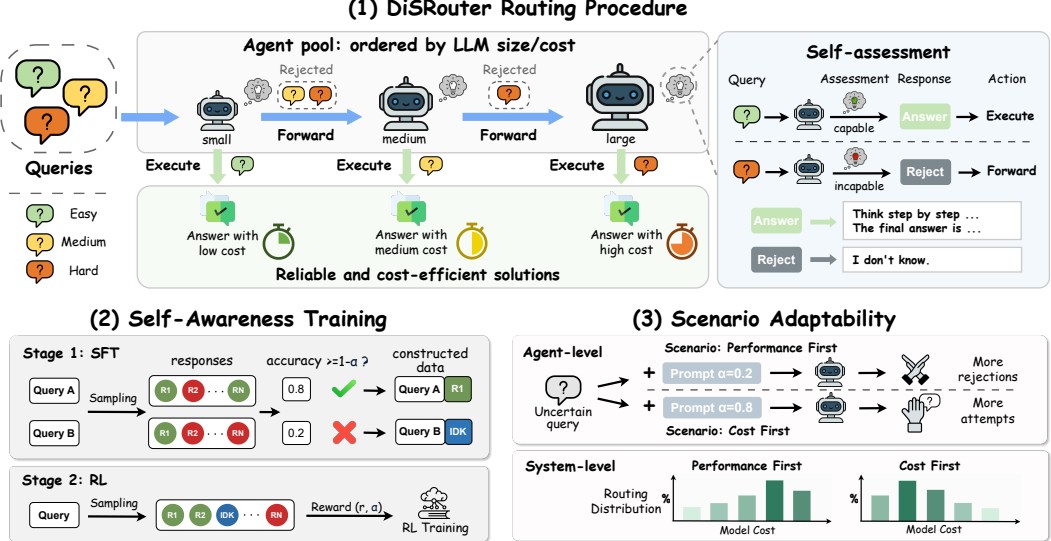

Figure 2: **The *DiSRouter* Framework. (1) The routing procedure of *DiSRouter*:** Queries of varying difficulty enter a routing cascade ordered by LLM size/cost, where each LLM agent uses self-assessment to decide whether to answer the query or reject it and route it to the next agent. **(2) The Self-Awareness Training pipeline**. Green and red indicate correct and incorrect answers, respectively, while blue represents the rejection behavior ("I don't know"). **(3) *DiSRouter* possesses scenario adaptability**: At the agent level, LLMs adjust their behavior toward uncertain queries based on scenario requirements. At the system level, the overall routing distribution shifts with the scenario: a greater emphasis on cost leads to more queries being routed to smaller models.

For any query $x$, the routing process begins at an entry-point agent, typically $m_1$. This generates a routing path, or trajectory, $P(x) = (m_{i_1}, m_{i_2}, \ldots, m_{i_k})$, where $i_1 = 1$. Each subsequent agent in the path is determined by the policy of the previous one, such that $m_{i_{j+1}} = \pi_{i_j}(x)$ for $j < k$. The path terminates until an agent $m_{i_k}$ decides to execute, i.e., $\pi_{i_k}(x) = m_{i_k}$. The final agent to process the query is thus denoted as $\Pi(x) = m_{i_k}$. Since routing costs are negligible, the total cost is simply the inference cost of the final model: $C(\Pi(x)) = C(m_{i_k}) = c_{i_k}$.

The objective is to find the optimal set of local policies $\Pi^*$ that maximizes the system utility:

$$\Pi^* = \arg\max_{\Pi} \ \mathbb{E}_{x \sim D}[A(\Pi(x), x) - \alpha \cdot C(\Pi(x))] \tag{5}$$

**Distributed Optimization**  A key advantage of the distributed architecture is that the complex systemic optimization problem can be decomposed, allowing each agent's routing policy $\pi_i$ to be optimized independently. The policies of different agents can vary, leveraging either the inherent self-knowledge of LLM (Yin et al., 2023) or a further aligned and improved self-awareness (Xu et al., 2024). Under this paradigm, each agent $m_i$ learns its optimal local policy $\pi_i^*$ by maximizing its own expected local utility $U_i$ based solely on locally available information, which can be formulated as follows, where the design of $U_i$ can be tailored to each individual agent.

$$\pi_i^* = \arg\max_{\pi_i} \ U_i(\pi_i) \tag{6}$$

This distributed optimization framework allows individual agents to be trained or updated without requiring a full-system retraining, enhancing the scalability and flexibility of the entire system.

## 3   DISROUTER

To overcome the limitations of conventional centralized routers, we propose *DiSRouter* (Distributed Self-Router), a framework built on a distributed architecture. It relies on the *self-awareness* of each

agent as its local routing policy, which we define as an agent's intrinsic ability to accurately assess if a query falls within its knowledge boundaries and can be solved correctly. Beyond the inherent advantages of a distributed structure, *DiSRouter* also possesses Scenario Adaptability, the ability to dynamically adjust its collective behavior based on scenario requirements, from Performance-First to Cost-First. This system-level capability is contingent on each agent being scenario-adaptive.

## 3.1 ROUTING PROCEDURE OF DiSROUTER

To facilitate a clear and focused evaluation, we adopt a simple cascade structure for *DiSRouter* in this work, with the routing procedure illustrated in Figure 2 (1). Agents are ordered by increasing cost, and each agent $m_i$'s action space is limited to either executing the query or routing it to the next agent, $m_{i+1}$. We realize this through a "reject" behavior: an agent answers a query if confident, otherwise it rejects and routes it to the next agent, which transforms the abstract notion of self-awareness into a concrete action. Consequently, *DiSRouter* shifts the paradigm from training a single, potentially limited router to training a set of distributed, reliable self-assessment policies. To this end, we propose the following Self-Awareness Training pipeline for LLM alignment.

## 3.2 SELF-AWARENESS TRAINING

The *DiSRouter* routing system hinges on the self-awareness of each agent. Although original LLMs possess some inherent self-knowledge, and certain uncertainty-based methods have been proposed to evaluate LLM output confidence (Huang et al., 2024), they often struggle to accurately assess an LLM's capacity for specific queries (Zheng et al., 2025). It is worth noting that high-capability closed-source LLMs (e.g., GPT-4) may already possess strong intrinsic self-awareness and can be integrated without additional training, as discussed in Appendix A.9. To address the limitations of open-source LLMs, we introduce a two-stage training pipeline to augment LLM **self-awareness**, comprising an SFT (Supervised Fine-Tuning) stage for foundational self-assessment abilities, followed by an RL (Reinforcement Learning) stage for further improvement. The data construction process is illustrated in Figure 2 (2). Beyond self-awareness, we also train agents to be **scenario-adaptive**: they should be more conservative in a "Performance First" scenario to minimize errors, and more aggressive in a "Cost First" scenario for greater efficiency. The scenario adaptability of *DiSRouter* is shown in Figure 2 (3).

### 3.2.1 SCENARIO ADAPTABILITY INSTRUCTIONS

For a given query, the current scenario's requirements are conveyed to the LLMs via instructions. We add instructions corresponding to different scenarios into the prompts, explicitly conveying user preferences to enable nuanced behavioral adjustments. For detailed prompts, see Appendix A.12.1.

### 3.2.2 SELF-AWARENESS SFT

We adopt a data construction method similar to prior work on improving LLM reliability (Xu et al., 2024) by introducing explicit rejection behavior: models are trained to respond with "I don't know" when uncertain. We use the target LLM to perform $N$ CoT (Chain-of-Thought) inference trials per training query, with the frequency of correct answers measuring its capability for each query. Queries with a correctness frequency below a certain threshold $\delta$ are deemed unanswerable, and their responses are replaced with the rejection statement: "I don't know." For the remaining high-capability queries, which the model is expected to answer, their thought processes and answers are extracted and standardized into an "Answer" template (see Appendix A.12.2).

To accommodate different scenarios with varying preference factors $\alpha$, we set the rejection threshold $\delta$ as $1 - \alpha$. For instance, in a "Performance First" scenario ($\alpha = 0.2$), the model should only answer when highly confident ($\delta = 0.8$). We construct an equal amount of training data for three distinct scenarios and mix them during training, while maintaining an equal proportion of accepted and rejected samples to prevent bias.

### 3.2.3 SELF-AWARENESS RL

To further enhancing self-awareness and scenario adaptability, we propose a scenario-conditioned reward for RL. For a given query $x$, the reward is:

$$\text{reward}(x) = \begin{cases} 1, & \text{if answer correctly} \\ 0, & \text{if answer incorrectly} \\ (1-\alpha)^{\gamma}, & \text{if reject} \end{cases} \tag{7}$$

Here, $\alpha$ is the preference factor that represents the scenario's emphasis on cost, with a larger $\alpha$ indicating greater cost importance. $\gamma$ is the reliability factor, introduced to ensure the model maintains sufficient reliability and does not excessively sacrifice precision for cost. Both parameters range from 0 to 1. For RL training, we maintain an equal proportion of data from all three scenarios, which are then combined to form the final training dataset.

**Reward Rationality Analysis** A model's decision to answer query $x$ or not is essentially based on whether the expected reward for answering is greater than the expected reward for rejecting. Let these be $E[\text{answer}]$ and $E[\text{reject}]$, and $p(x)$ denote the model's capability in answering $x$. Then:

$$E[\text{answer}] = [p(x) \cdot 1 + (1 - p(x)) \cdot 0] = p(x) \tag{8}$$

$$E[\text{reject}] = (1-\alpha)^{\gamma} \tag{9}$$

$$E[\text{answer}] > E[\text{reject}] \Rightarrow p(x) > (1-\alpha)^{\gamma} \tag{10}$$

This implies that an LLM's capability on query $x$ must reach a certain threshold before it chooses to answer. This threshold decreases as $\alpha$ increases, meaning that the model prioritizes cost more, its capability threshold for answering decreases, making the LLM more aggressive. This behavior aligns with our requirements. If the reliability factor $\gamma = 1$, this trend is linear. However, the human preference for accuracy and cost is often nonlinear; we typically desire the system to maintain high reliability most of the time, only allowing a significant decrease in accuracy when extreme cost-saving is prioritized. The graph of the function $f(\alpha) = (1-\alpha)^{\gamma}$ is shown in Figure 3. Thus, using a $\gamma$ value between 0 and 1 can effectively represent this demand. In this work, we use $\gamma = 0.5$ to reflect our pursuit of high reliability.

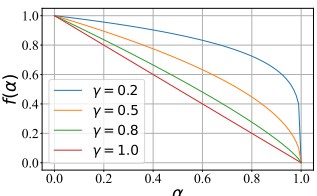

Figure 3: $f(\alpha) = (1-\alpha)^{\gamma}$

## 4 EXPERIMENTS

### 4.1 EXPERIMENTAL SETUP

**Datasets** We utilize widely used benchmarks covering various domains as our test tasks, including mathematical reasoning, common sense question answering, and reading comprehension. These comprise seven datasets: **GSM8K** (Cobbe et al., 2021), **AI2 Reasoning Challenge (ARC)** (Clark et al., 2018), **MMLU** (Hendrycks et al., 2020), **RACE_HIGH** (Lai et al., 2017), **OpenbookQA** (Mihaylov et al., 2018), **DROP** (Dua et al., 2019), and **CosmosQA** (Huang et al., 2019). These datasets serve as in-domain data for model training and evaluation. Additionally, we incorporate several datasets as out-of-domain tasks to assess the generalization capability of the various methods, including: **SQuAD** (Rajpurkar et al., 2018), **HellaSwag** (Zellers et al., 2019), and **HeadQA** (Vilares & Gómez-Rodríguez, 2019). For datasets with a hidden test set, we utilize the validation set as an alternative. We evaluate LLMs in a zero-shot Chain-of-Thought (CoT, Wei et al., 2022) manner and use accuracy as the evaluation metric for all these datasets. Further details on dataset descriptions, statistics, and prompts for testing can be found in Appendix A.2.

**LLM Pool** Our model pool comprises five distinct sizes of the Qwen2.5-Instruct series, spanning from 0.5B to 14B (Qwen et al., 2025). These LLMs vary significantly in terms of performance and inference cost. Following GraphRouter (Feng et al., 2024), we assign each LLM a normalized, fixed cost ranging from 0 to 1. Table 1 summarizes their average performance across datasets and corresponding costs. Details of LLM performance and actual inference time on every dataset are provided in Appendix A.3.

Table 1: **Performance and cost for LLMs of different sizes.** ID: in-domain, OOD: out-of-domain.

| Model Size | ID Accuracy | OOD Accuracy | Model Cost |
|---|---|---|---|
| 0.5B | 38.01 | 35.82 | 0.1 |
| 1.5B | 58.95 | 54.69 | 0.2 |
| 3B | 73.08 | 61.57 | 0.4 |
| 7B | 80.90 | 69.89 | 0.7 |
| 14B | 85.45 | 74.43 | 0.9 |

Table 2: **Comparison of different methods on in-domain datasets across three scenarios.** We calculated and report the average values across multiple datasets.

| Method | | Performance First: $\alpha = 0.2$ | | | Balance: $\alpha = 0.5$ | | | Cost First: $\alpha = 0.8$ | | |
|---|---|---|---|---|---|---|---|---|---|---|
| | | Accuracy ↑ | Cost ↓ | Utility ↑ | Accuracy ↑ | Cost ↓ | Utility ↑ | Accuracy ↑ | Cost ↓ | Utility ↑ |
| Topline | Oracle | 0.93 | 0.29 | 0.87 | 0.93 | 0.29 | 0.79 | 0.93 | 0.29 | 0.70 |
| Naive Baselines | Smallest LLM | 0.38 | 0.10 | 0.36 | 0.38 | 0.10 | 0.33 | 0.38 | 0.10 | 0.30 |
| | Largest LLM | 0.85 | 0.90 | 0.67 | 0.85 | 0.90 | 0.40 | 0.85 | 0.90 | 0.13 |
| | Random | 0.67 | 0.46 | 0.58 | 0.67 | 0.46 | 0.44 | 0.67 | 0.46 | 0.30 |
| Router Baselines | RouteLLM | 0.44 | 0.16 | 0.41 | 0.44 | 0.16 | 0.36 | 0.44 | 0.16 | 0.31 |
| | FrugalGPT | 0.70 | 0.32 | 0.64 | 0.63 | 0.22 | 0.52 | 0.60 | 0.21 | 0.43 |
| | Automix | 0.85 | 1.10 | 0.63 | 0.58 | 0.58 | 0.29 | 0.44 | 0.33 | 0.18 |
| | FORC | 0.78 | 0.65 | 0.65 | 0.69 | 0.45 | 0.47 | 0.60 | 0.29 | 0.37 |
| | GraphRouter | 0.82 | 0.68 | 0.68 | 0.77 | 0.46 | 0.53 | 0.61 | 0.20 | 0.45 |
| **DiSRouter (Ours)** | SFT | 0.84 | 0.59 | 0.72 | 0.83 | 0.52 | 0.57 | 0.77 | 0.42 | 0.43 |
| | + RL | 0.81 | 0.40 | **0.73** | 0.77 | 0.32 | **0.61** | 0.75 | 0.29 | **0.52** |

**Baseline Methods** We compare our proposed method against the following baselines. We first introduce three intuitive Naive Baselines: (1) **Largest** / (2) **Smallest LLM**: Always routes queries to the largest / smallest available LLM. (3) **Random**: Randomly routes each query to an LLM.

We also incorporate several representative routing methods: (4) **RouteLLM** (Ong et al., 2024), (5) **FrugalGPT** (Chen et al., 2023), (6) **Automix** (Aggarwal et al., 2024), (7) **FORC** (Šakota et al., 2024), and (8) **GraphRouter** (Feng et al., 2024). Details on the description and implementation of these methods can be found in Appendix A.4. Notably, Automix relies on the self-verification of the LLMs, and we consider the time overhead for this process equivalent to the LLM's response cost.

Additionally, we introduce a theoretically optimal routing strategy (9) **Oracle**, as a topline for comparative analysis, which always routes queries to the smallest capable LLM.

**Metrics** We compare the average accuracy and average cost of each routing system across multiple datasets. To evaluate the overall routing performance, we utilize the utility metric introduced in §2.1, Equation (2), which can be adjusted for different scenarios by setting a corresponding value for $\alpha$.

**Implementation Details** For Self-Awareness SFT, we randomly selected 10,000 training samples from all in-domain datasets and trained for one epoch. The learning rate was set to $1 \times 10^{-5}$, using four A800 GPUs with a batch size of four per GPU. For Self-Awareness RL, we employed the Verl (Sheng et al., 2024) framework with the Reinforce++ algorithm. We used 10,000 training samples with a total batch size of 128. The learning rate was $1 \times 10^{-6}$, and each query was sampled eight times with a temperature $T$ of 1. Training was conducted for one epoch on eight A800 GPUs.

### 4.2 RESULTS AND ANALYSIS

Table 2 presents the performance of *DiSRouter* on in-domain datasets compared to various baselines. Across all three scenarios, our method consistently achieves the highest utility, outperforming all baselines. Notably, *DiSRouter* reaches at least 74.29% of the Oracle topline, demonstrating the effectiveness of our distributed routing approach. Compared to Naive Baselines, our system is better at balancing performance and cost according to the demands of each scenario. Furthermore, our method generally achieves higher accuracy with lower costs than existing Router Baselines. A more detailed analysis of why *DiSRouter* routes better is provided in Section 4.3.

**Generalization Analysis** We evaluate the generalization capabilities of *DiSRouter* from three critical dimensions: task domains, model architectures, and task types. First, regarding **task domains**, we compare the performance of different baselines and *DiSRouter* on several out-of-domain datasets (Table 3). Our findings indicate that *DiSRouter* effectively distinguishes whether a query is within its capability even on unseen datasets (see Appendix A.5 for full results). Second, regarding **model architecture**, we validate our framework on a heterogeneous agent pool comprising diverse LLM families (e.g., Gemma-3 (Team et al., 2025), Phi-4 (Abdin et al., 2024)). As detailed in Appendix A.10, results demonstrate that our training pipeline and distributed architecture remain highly effective across different LLM architectures. Third, regarding **task types**, we extend our evaluation to open-ended generative tasks (e.g., text summarization) where correctness is not binary. As shown in

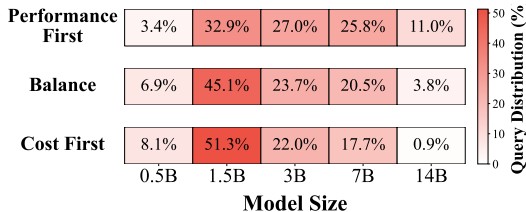

Figure 4: **System-level routing distribution.**

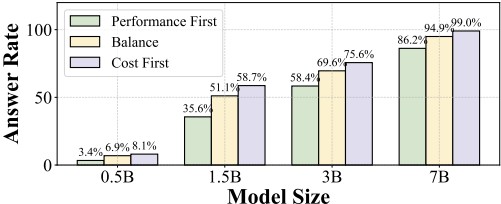

Figure 5: **Agent-level answer rate.**

Appendix A.11, by adapting the reward formulation to continuous metrics, *DiSRouter* successfully generalizes to these scenarios, distinguishing high-quality generations from low-quality ones.

**Modularity and Flexibility Analysis** The distributed structure of *DiSRouter* inherently offers "plug-and-play" modularity. To validate this, we reduced the agent pool from the original five models to a three-agent system (1.5B, 3B, and 14B), without the need for any retraining or modification to the agents. Results for the "Balance" scenario are presented in Table 4 (see Appendix A.6 for full results). Our three-agent *DiSRouter* system still achieved superior utility across all three scenarios, confirming the excellent modularity of *DiSRouter*. Furthermore, the modest utility drop compared to the five-agent system was expected, indicating that our full system effectively leverages the two additional models for performance gains. This scaling of system performance with the diversity of available agents further validates the soundness of our distributed, self-routing paradigm.

Table 3: **Comparison of different methods on out-of-domain datasets for Balance scenario.**

| Method | | Balance: $\alpha = 0.5$ | | |
|---|---|---|---|---|
| | | Accuracy ↑ | Cost ↓ | Utility ↑ |
| Topline | Oracle | 0.89 | 0.33 | 0.72 |
| Naive Baselines | Smallest LLM | 0.36 | 0.10 | 0.31 |
| | Largest LLM | 0.74 | 0.90 | 0.29 |
| | Random | 0.59 | 0.46 | 0.36 |
| Router Baselines | RouteLLM | 0.42 | 0.22 | 0.31 |
| | FrugalGPT | 0.55 | 0.25 | 0.42 |
| | Automix | 0.55 | 0.60 | 0.25 |
| | FORC | 0.63 | 0.53 | 0.36 |
| | GraphRouter | 0.64 | 0.49 | 0.39 |
| DiSRouter (Ours) | SFT | 0.74 | 0.62 | 0.43 |
| | + RL | 0.69 | 0.43 | **0.48** |

Table 4: **Performance comparison on a reduced 3-agent system for Balance scenario.** RouteLLM and Automix are excluded as they require retraining for agent pool modifications.

| Method | | Balance: $\alpha = 0.5$ | | |
|---|---|---|---|---|
| | | Accuracy ↑ | Cost ↓ | Utility ↑ |
| Topline | Oracle | 0.91 | 0.38 | 0.72 |
| Naive Baselines | Smallest LLM | 0.59 | 0.20 | 0.49 |
| | Largest LLM | 0.85 | 0.90 | 0.40 |
| | Random | 0.72 | 0.50 | 0.47 |
| Router Baselines | FrugalGPT | 0.63 | 0.22 | 0.51 |
| | FORC | 0.72 | 0.48 | 0.48 |
| | GraphRouter | 0.75 | 0.46 | 0.52 |
| DiSRouter (Ours) | SFT | 0.83 | 0.54 | 0.56 |
| | + RL | 0.78 | 0.36 | **0.60** |

**Scenario Adaptability Analysis** *DiSRouter* dynamically adjusts its collective behavior via the preference factor $\alpha$ across different scenarios, a capability we term Scenario Adaptability. Crucially, unlike baselines like GraphRouter that are scenario-dependent and require re-training for each scenario, *DiSRouter* is a single, unified system that adapts to diverse scenarios using a scenario-specific prompt. We analyze this capability at both the system and the individual agent levels.

At the system level, figure 4 shows a clear strategic shift in the distribution of queries routed to each agent. As the scenario transitions from "Performance First" to "Cost First", more queries are routed to smaller models, indicating that *DiSRouter* shifts its strategy from maximizing accuracy to prioritizing cost. This systemic adjustment stems from the adaptability of each individual agent. At the agent level, we plot the answer rate—the percentage of queries an agent chooses to answer rather than reject—for each agent across three scenarios in Figure 5. As $\alpha$ increases towards a cost-first priority, every agent's decision threshold is effectively lowered, making them more "willing" to answer. This synchronized adaptation is the fundamental mechanism enabling the system to achieve global objectives without explicit coordination, confirming that our localized, scenario-conditioned reward function successfully instills the desired Scenario Adaptability in each agent.

### 4.3 WHY DiSROUTER ROUTES BETTER?

In this section, we explore the reasons for *DiSRouter*'s superior routing utility. We first show that *DiSRouter* is better at distinguishing between easy and hard queries than router baselines, and after that we analyze the self-awareness of our trained LLMs, revealing their strong intrinsic capacity to determine whether a problem falls within their capabilities and can be answered correctly. It is important to note that *DiSRouter*'s performance gains do not stem from a direct improvement in task performance due to training, a point we validate with experiments in Appendix A.8.

**System-level Analysis**   We define queries solvable by 3B and smaller models as "Easy," and all others as "Hard." An effective routing system should direct easier queries to smaller LLMs for lower average costs. Under the "Balance" scenario, we analyzed the average cost for easy and hard queries for three baselines—GraphRouter, FORC, and FrugalGPT—compared to our method (Figure 6). It is evident that the router baselines lack or exhibit only limited discriminative capability. Our system, in contrast, effectively differentiates queries of varying difficulty.

**External Router vs. Intrinsic Assessment**
A core advantage of *DiSRouter* is its reliance on intrinsic self-assessment over an external router. To demonstrate this, we formalize the question "whether a LLM can solve query $x$" as a binary classification problem. We targeted the 7B model, using the correctness of its greedy-search response on each query as ground truth. To ensure a balanced test set, we randomly sampled 5,000 instances from each

Table 5: **Comparison between external routers and self-assessment of *DiSRouter*.**

| Type | Classifier | Size | Accuracy | F1 |
|------|-----------|------|----------|-----|
| Naive | Random | - | 0.50 | 0.50 |
| External | Bert-based | 0.13B | 0.56 | 0.63 |
|  | LLM-based | 8B | 0.71 | 0.77 |
| Intrinsic | DiSRouter | 7B | 0.80 | 0.81 |

class. We trained two external classifiers (one BERT-based, one LLM-based using Llama3-8B-Instruct (Grattafiori et al., 2024)), and compared their performance against our model. For our *DiSRouter*-aligned LLM, its decision whether to reject a query serves as the classification output. Implementation details for classifiers are in Appendix A.7, with results summarized in Table 5.

Our self-assessment method demonstrates superior classification accuracy and F1 score, indicating its ability to accurately discern solvable queries and prevent itself from queries beyond its capabilities. External routers exhibit two primary limitations: (1) **Limited Performance due to Constrained Parameters.** While router performance improves with larger models, using a separate, larger router introduces significant overhead, a problem *DiSRouter* avoids by integrating the rejection action. (2) **Inferiority of External Evaluation.** Even a similar-sized LLM-based external classifier lags behind our intrinsic approach. We therefore argue that integrating the assessment function directly into the LLM, as *DiSRouter* does, is a more rational and efficient solution.

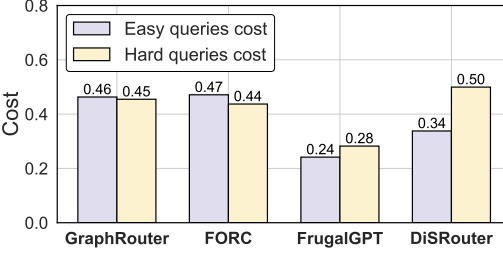

Figure 6: **Comparison of average Cost of easy and hard queries for "Balance" scenario.** A larger discrepancy between the two indicates a stronger ability to differentiate query difficulty.

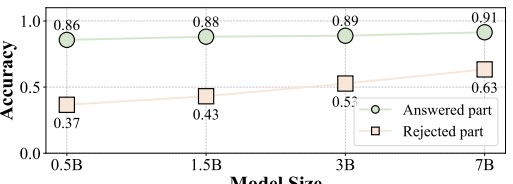

Figure 7: **Accuracy of answered and rejected queries for *DiSRouter*-aligned LLMs in the "Performance First" scenario.** The accuracy of our models on answered queries significantly surpasses that of their rejected counterparts, providing evidence of effective self-assessment.

**Self-Awareness Ability**   We further analyzed the self-awareness of each aligned LLM based on their query acceptance and rejection behaviors. Figure 7 compares their accuracy on answered versus rejected queries under the "Balance" scenario. For the rejected queries, we employed a prefix injection approach (Li et al., 2025) by appending an affirmative prefix to the end of the prompt.

This forced the models to generate responses, enabling us to calculate their underlying accuracy. It is evident that our aligned models possess a strong self-awareness capability, demonstrating a significant capacity to distinguish whether a problem falls within their capabilities.

These findings validate our core motivation: methods that rely on a centralized external router to assess query difficulty for routing decisions are inherently limited. Such approaches primarily learn to route queries to LLMs with higher expected utility rather than making judgments based on the actual difficulty of the query. Our *DiSRouter* framework, which leverages the models' self-awareness ability, is not only logically more sound but has also been proven to be more effective.

## 5 RELATED WORK

Research in LLM query routing is largely dominated by centralized architectures. Early systems like FrugalGPT (Chen et al., 2023) used a scoring model to directly evaluate the quality of an LLM response to decide whether to proceed to a more powerful agent. More recent methods focus on predictive routing, where a central router assesses queries before execution. These routers typically predict either the difficulty of a query (e.g., Hybrid LLM (Ding et al., 2024), RouteLLM (Ong et al., 2024)) or the expected performance of each available LLM, using techniques like bandit-based models (e.g., C2MAB-V (Dai et al., 2024)) or graph neural networks (e.g., GraphRouter (Feng et al., 2024)). To address the inherent rigidity of centralized networks, recent works have explored multi-dimensional profiling (Shi et al., 2025) and decoupled regression predictors (Wang et al., 2025). While effective, these systems share fundamental limitations: their overall performance is bottlenecked by the capability of the external router, and their centralized architectures are often rigid, lacking the modularity to easily adapt to a dynamically changing pool of agents.

To bypass the bottlenecks of external classifiers, alternative approaches explore the intrinsic knowledge of LLMs for routing. For instance, AutoMix (Aggarwal et al., 2024) utilizes an LLM's self-verification to guide routing decisions; however, its multi-step verification incurs significant time overhead and relies on static, non-adaptive thresholds. To mitigate such latency, methods like Early Abstention cascades (Michael et al., 2025) trigger routing decisions at early decoding stages based on error probability distributions. Moving towards fully distributed topologies, AgentNet (Yang et al., 2025) shares a similar motivation with our work by equipping each agent with an independent router for decentralized decision-making.

## 6 DISCUSSION AND CONCLUSION

**Discussion on Distributed Routing Networks**   While our work mainly discusses a simple cascading structure, the distributed nature of *DiSRouter* theoretically supports more complex network topologies, such as cross-level cascading or tree-like structures with multiple routing directions. To implement these, we would need to introduce system-level information during the inference phase, a practice common in distributed communication (e.g., Content Distribution Network, CDN). This would allow each model to be aware of subsequent models' information when making routing decisions, enabling more efficient routing, such as directing the most difficult queries to the final model.

**Future Work**   We acknowledge several avenues for future research. (1) Our current Self-Awareness Training pipeline is relatively basic. Future work could explore incorporating "reasoned refusals" or more sophisticated distributed reinforcement learning (RL) rewards to further enhance model self-awareness, especially for smaller models. (2) We will explore more complex distributed network structures and implement the system information exchange during inference to generalize the framework to a true intelligent agent network.

**Conclusion**   This paper introduces the Distributed Self-Router (*DiSRouter*), a novel framework designed to address the challenge of cost-effective query routing in multi-agent systems. By identifying key limitations in prevailing centralized routing architectures, *DiSRouter* eliminates the central router and instead empowers each LLM agent with intrinsic self-awareness, creating a fully distributed, scalable, and plug-and-play system. Our extensive experiments validate the superiority of *DiSRouter*, which consistently achieves higher utility than competitive baselines across various cost-sensitive scenarios. A key advantage of *DiSRouter* is its reliance on intrinsic self-awareness, which our experiments prove outperforms external evaluation.

## ACKNOWLEDGMENTS

This work is funded by the China NSFC Projects (62120106006, 62576212, 92370206, and U23B2057) and Shanghai Municipal Science and Technology Project (25X010202846).

## ETHICS STATEMENT

The authors declare that this work was conducted in full accordance with the released code of ethics. We confirm that there are no ethical issues to be disclosed.

## REPRODUCIBILITY STATEMENT

We are committed to the principles of reproducible research. Our work is highly reproducible as all components are based on publicly available resources and are described in detail throughout this paper. The models and datasets utilized in this study are open-source, with further details provided in Section 4.1. A comprehensive description of our data construction process is presented in Section 3.2. Furthermore, all implementation details, including training parameters and experimental settings, are thoroughly documented in Section 4.1. We believe that these provisions will enable the research community to fully replicate our findings.

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

# Appendix

TABLE OF CONTENTS

## THE USE OF LARGE LANGUAGE MODELS (LLMS)

In the preparation of this paper, we utilized Large Language Models (LLMs) as an auxiliary tool to support our research and writing process. Specifically, LLMs were employed for two primary purposes: 1) to aid in the discovery of relevant literature and assist in the paper reading by generating summaries, and 2) to aid in manuscript refinement through the translation of specific phrases and the polishing of sentence structures for improved clarity and readability. We affirm that the authors critically reviewed, revised, and edited all content suggested by the LLMs, and we take full responsibility for the accuracy and integrity of the entire work.

### A.1 TIME OVERHEAD ANALYSIS

As discussed in Section 2.1, the total cost of a routing system comprises the routing cost (the time overhead of the decision process) and the inference cost (the execution time of the selected LLM). In this section, we conduct a fine-grained statistical analysis of the actual time consumption. To accurately measure the inference time for each sample, we excluded parallel acceleration techniques typically provided by inference engines like vLLM (Kwon et al., 2023). Instead, we utilized a sequential, sample-by-sample inference approach for precise timing.

Table 6: **Comparison of average time consumed by rejection and answering responses.** Statistics were gathered using predictions from the base model (for the 14B LLM) and from SFT-trained models (for the others). Proportion refers to the proportion of time consumption of rejection to answering.

| Type | 0.5B | 1.5B | 3B | 7B | 14B |
|---|---|---|---|---|---|
| Answer (s) | 0.825 | 1.399 | 2.446 | 3.895 | 5.398 |
| Reject (s) | 0.039 | 0.053 | 0.066 | 0.083 | - |
| Proportion (%) | 4.7 | 3.8 | 2.7 | 2.1 | - |

Table 7: **Analysis of routing overhead in *DiSRouter*.** We report the time consumption for both average and worst-case (routing to the final 14B model) situations under "Balance" scenario. Proportion refers to the proportion of Routing Cost to Inference Cost.

| | Inference Cost (s) | Routing Cost (s) | Proportion (%) |
|---|---|---|---|
| Average | 2.270 | 0.081 | 3.56 |
| Worst Case | 5.398 | 0.241 | 4.46 |

***DiSRouter* Overhead Analysis** In the *DiSRouter* framework, the routing decision is intrinsic, manifested as the time taken by an agent to generate a "rejection" response. Therefore, the routing cost corresponds to the cumulative time consumed by all agents in the routing path to generate their rejections. Our empirical analysis shows that the average rejection time for a Self-Awareness trained model is less than 5% of its inference time (Table 6). Besides, we further analyzed both the average routing overhead across all samples and the worst-case scenario (where the query traverses the entire cascade to the final 14B model). As shown in Table 7, the routing overhead constitutes less than 5% of the total inference time in both average and worst-case scenarios. Consequently, we consider this routing cost negligible and do not factor it into the utility calculations in the main text.

Table 8: **Comparison of System Latency (Time to First Token, s) across different methods.**

| | RouteLLM | FrugalGPT | Automix | FORC | GraphRouter | *DiSRouter* |
|---|---|---|---|---|---|---|
| Average | 0.048 | 2.620 | 3.475 | 0.070 | 0.024 | 0.081 |
| Worst Case | 0.063 | 9.565 | 11.34 | 0.076 | 0.028 | 0.241 |

**System Latency Comparison**   To evaluate the real-time responsiveness of different systems, we benchmarked the System Latency across different methods, which is defined as the time required to generate the first token of the final answer. we measured the time consumption of router baselines under a similar sample-by-sample configuration, using 100 randomly sampled test queries. The comparison is presented in Table 8. The results reveal distinct characteristics among routing strategies.

1. **Predictive routers** utilizing lightweight external classifiers (RouteLLM, FORC and GraphRouter) exhibit the lowest latency, as their routing decisions are nearly instantaneous.

2. **Generative/Cascading routers** that rely on intermediate model outputs (FrugalGPT) or extensive self-verification steps (AutoMix) incur significant latency, often exceeding the inference time of the LLMs themselves.

3. *DiSRouter* demonstrates a balanced profile. While its latency is marginally higher than that of predictive routers due to the token generation process for rejection, it is significantly faster than other cascading or verification-based approaches. Given that the rejection response typically consists of only a few tokens, **the introduced overhead remains minimal and does not negatively impact the user experience.**

## A.2   DATASET DESCRIPTIONS

This section provides a brief overview of the datasets utilized in this work.

**GSM8K**: A dataset focused on mathematical problem-solving, requiring models to comprehend natural language problems, apply correct operations, and provide solutions.

**ARC**: Designed to evaluate models' ability to answer questions based on scientific texts, involving comprehension, critical thinking, and scientific reasoning.

**MMLU**: A comprehensive dataset testing broad knowledge and reasoning across various domains, featuring multiple-choice questions from diverse subjects.

**RACE_HIGH**: A reading comprehension benchmark for high school-level texts, testing deep comprehension, logical reasoning, and inference skills.

**OpenbookQA**: Crafted to test a model's ability to apply elementary science knowledge to novel situations, requiring integration of facts with commonsense reasoning.

**DROP**: A challenging dataset evaluating a system's ability to perform discrete operations over text, such as addition or counting, on adversarially created questions.

**CosmosQA**: Focuses on commonsense-based reading comprehension, requiring models to infer implicit causes, effects, intentions, or plausible outcomes from narratives.

**SQuAD**: An extractive reading comprehension benchmark where the answer is a text span directly found within a corresponding Wikipedia passage.

**HellaSwag**: A dataset for commonsense reasoning tasks, where models predict the most plausible continuation for diverse incomplete situations.

**HeadQA**: Evaluates multilingual and multi-domain question answering in healthcare, featuring multiple-choice questions from real medical exams in Spanish and English.

Statistical information for these datasets and the prompts for testing are summarized and presented in Table 11.

## A.3   LLM POOL PERFORMANCE AND COST

We present the performance and actual computational costs of the five different-sized models from the Qwen2.5-Instruct series on our selected datasets in tables 9 and 10, respectively.

Table 9: **Performance of LLMs (accuracy%).**

| Dataset | 0.5B | 1.5B | 3B | 7B | 14B |
|---|---|---|---|---|---|
| GSM8K | 42.76 | 68.31 | 84.00 | 86.81 | 91.58 |
| ARC | 37.71 | 62.71 | 75.51 | 83.02 | 85.58 |
| MMLU | 34.97 | 55.19 | 63.77 | 74.04 | 79.21 |
| RACE_HIGH | 43.57 | 63.69 | 77.67 | 85.79 | 89.14 |
| OpenbookQA | 42.40 | 66.40 | 81.20 | 87.80 | 94.2 |
| DROP | 28.82 | 43.37 | 62.72 | 70.37 | 73.25 |
| CosmosQA | 35.81 | 53.00 | 66.67 | 78.46 | 85.19 |
| SQUAD | 44.48 | 62.87 | 61.17 | 73.63 | 71.76 |
| HellaSwag | 29.91 | 48.71 | 62.24 | 63.91 | 72.78 |
| HeadQA | 33.08 | 52.48 | 61.31 | 72.14 | 78.74 |
| Average | 37.35 | 57.67 | 69.62 | 77.6 | 82.14 |

Table 10: **Cost of LLMs (inference time, s).**

| Dataset | 0.5B | 1.5B | 3B | 7B | 14B |
|---|---|---|---|---|---|
| GSM8K | 0.79 | 1.48 | 2.35 | 3.62 | 4.70 |
| ARC | 0.79 | 1.31 | 2.26 | 3.65 | 4.90 |
| MMLU | 0.72 | 1.28 | 1.72 | 3.11 | 4.83 |
| RACE_HIGH | 0.73 | 1.64 | 2.76 | 4.15 | 5.90 |
| OpenbookQA | 0.43 | 1.09 | 1.40 | 3.62 | 4.98 |
| DROP | 0.72 | 2.45 | 3.40 | 6.11 | 7.98 |
| CosmosQA | 0.78 | 1.85 | 2.85 | 5.07 | 5.71 |
| SQUAD | 0.62 | 1.73 | 3.37 | 4.86 | 5.95 |
| HellaSwag | 0.55 | 1.15 | 2.78 | 4.28 | 5.31 |
| HeadQA | 0.63 | 0.73 | 2.54 | 3.92 | 3.69 |
| Average | 0.68 | 1.47 | 2.54 | 4.24 | 5.40 |

Table 11: Statistical information and CoT prompts for datasets.

| Dataset | Train samples | Test samples | CoT prompt |
|---|---|---|---|
| GSM8K | 7,473 | 1,319 | Given the following math problem, reason and give a final answer to the problem. |
| ARC | 1,119 | 1,172 | Choose the most reasonable answer based on scientific reasoning and give a final answer. The final answer must be one of: [A, B, C, D] without any prefix or suffix. |
| MMLU | 4,993 | 14,042 | Choose the most reasonable answer of the multiple choice question based on scientific reasoning and give a final answer. The final answer must be one of: [A, B, C, D] without any prefix or suffix. |
| RACE_HIGH | 6,245 | 3,498 | Read the following passage carefully, choose the most suitable answer for the multiple-choice question based on commonsense reasoning and give a final answer. The final answer must be one of: [A, B, C, D] without any prefix or suffix. |
| OpenbookQA | 4,957 | 500 | Use both common sense and basic science knowledge to choose the best answer to the multiple-choice question. The final answer must be the one of: [A, B, C, D] without any prefix or suffix. |
| DROP | 15,480 | 9,535 | Read the following passage carefully and answer the question. You should give the final answer without any prefix or suffix. |
| CosmosQA | 25,262 | 2,985 | Read the given passage and choose the most plausible answer to the question based on context and common sense reasoning. The final answer must be one of: [A, B, C, D] without any prefix or suffix. |
| SQuAD | 86,821 | 5,928 | Answer the questions based on the provided context. Your response must follow the format requirement. The final answer must be the direct answer to the problem without any prefix or suffix. |
| HellaSwag | 39,905 | 10,042 | Choose the most reasonable continuation for the given text, reason and give a final answer. The final answer must be one of: [A, B, C, D] without any prefix or suffix. |
| HeadQA | 2,657 | 2,742 | Choose the most reasonable answer of the multiple choice question based on scientific reasoning and give a final answer. The final answer must be one of: [A, B, C, D] without any prefix or suffix. |

## A.4 ROUTER BASELINES: DESCRIPTIONS AND IMPLEMENTATION DETAILS

This section details the router baselines incorporated into this study, along with their respective implementation specifics.

**RouteLLM** Ong et al. (2024): This method frames the routing problem as a classification task, employing a Bert-style classifier to assign each request to the LLM that yields the highest reward. In the context of this paper, this translates to routing the query to the smallest model capable of resolving the problem. For our implementation, we utilized the base version of the RoBERTa model

as the backbone for the classifier. Training was conducted with a batch size of 128 under various learning rate configurations, selecting the model that performed best on the validation set (with a learning rate of $5 \times 10^{-5}$).

**FrugalGPT** Chen et al. (2023): This approach adopts a cascading strategy, where queries are progressively routed from smaller to larger models. A score model is trained to assess the output reliability of each model until a pre-determined, empirically searched threshold is exceeded. While the original paper employed a trained DistilBERT as the score model, we observed poor performance in our initial trials. Consequently, we opted for a higher-performing trained Qwen2.5-0.5B-Instruct as a substitute.

**Automix** Aggarwal et al. (2024): Also employing a cascading mechanism, Automix relies on the self-verification results generated after each model's response. It models the routing problem as a decision process and solves it using a Partially Observable Markov Decision Process (POMDP). The model undergoes ten reflection steps for each query and response. The number of bins for the self-verification probabilities within the POMDP instance was also set to 10. We configured the self-verification cost of the model to be equal to its response inference cost.

**FORC** Šakota et al. (2024): This method utilizes a trained meta-model to predict the score of each LLM on a given query, subsequently routing the query to the LLM possessing the maximum predicted utility. Similar to FrugalGPT, the original paper specified a trained DistilBERT as the score model. We replaced this with a more capable trained Qwen2.5-0.5B-Instruct. Training parameters for this substitute model were as follows: a maximum sequence length of 1024, a batch size of 16, a learning rate of $2 \times 10^{-5}$, and training for 10 epochs.

**GraphRouter** Feng et al. (2024): This approach incorporates textual descriptions related to tasks and LLMs as context, employing a graph neural network to predict the effect and cost of LLMs on queries. Queries are then routed to the LLM with the maximum predicted utility. We trained this model using the officially released code, modifying the reward calculation formula to align with our defined utility. We extensively searched for the optimal learning rate and training mask rate parameters, which are as follows for each scenario:

- **Performance First**: learning rate ($lr$) = $1 \times 10^{-4}$, training mask rate ($tmr$) = 0.1

- **Balance**: $lr = 2 \times 10^{-3}$, $tmr = 0.5$

- **Cost First**: $lr = 5 \times 10^{-3}$, $tmr = 0.7$

## A.5 OUT-OF-DOMAIN DATASETS DETAILED RESULTS

We have presented the results for out-of-domain datasets under the "Balance" scenario in the main text. Here, we present the results for the "Performance First" and "Cost First" scenarios, as shown in Table 12 and Table 13, respectively.

Table 12: **Comparison of methods on OOD datasets for Performance First scenario.**

| Method | | Performance First: $\alpha = 0.2$ | | |
| --- | --- | --- | --- | --- |
| | | Accuracy ↑ | Cost ↓ | Utility ↑ |
| Topline | Oracle | 0.89 | 0.33 | 0.82 |
| Naive Baselines | Smallest LLM | 0.36 | 0.10 | 0.34 |
| | Largest LLM | 0.74 | 0.90 | 0.56 |
| | Random | 0.59 | 0.46 | 0.50 |
| Router Baselines | RouteLLM | 0.42 | 0.22 | 0.38 |
| | FrugalGPT | 0.63 | 0.41 | 0.54 |
| | Automix | 0.74 | 1.10 | 0.52 |
| | FORC | 0.70 | 0.75 | 0.56 |
| | GraphRouter | 0.69 | 0.67 | 0.56 |
| **DiSRouter (Ours)** | SFT | 0.73 | 0.62 | 0.61 |
| | + RL | 0.75 | 0.67 | 0.62 |

Table 13: **Comparison of methods on OOD datasets for Cost First scenario.**

| Method | | Cost First: $\alpha = 0.8$ | | |
| --- | --- | --- | --- | --- |
| | | Accuracy ↑ | Cost ↓ | Utility ↑ |
| Topline | Oracle | 0.89 | 0.33 | 0.63 |
| Naive Baselines | Smallest LLM | 0.36 | 0.10 | 0.28 |
| | Largest LLM | 0.74 | 0.90 | 0.02 |
| | Random | 0.59 | 0.46 | 0.22 |
| Router Baselines | RouteLLM | 0.42 | 0.22 | 0.25 |
| | FrugalGPT | 0.55 | 0.21 | 0.38 |
| | Automix | 0.40 | 0.29 | 0.17 |
| | FORC | 0.54 | 0.32 | 0.29 |
| | GraphRouter | 0.53 | 0.20 | 0.37 |
| **DiSRouter (Ours)** | SFT | 0.69 | 0.56 | 0.25 |
| | + RL | 0.67 | 0.36 | 0.39 |

## A.6 REDUCED 3-AGENTS SYSTEM RESULTS

In Section 4.2, we validate the modularity of *DiSRouter* using a reduced 3-agent system. The results for the "Performance First" and "Cost First" scenarios are presented in Tables 14 and 15, respectively.

Table 14: **Comparison on a reduced 3-agent system for Performance First scenario.**

| Method | | Performance First: $\alpha = 0.2$ | | |
|---|---|---|---|---|
| | | Accuracy ↑ | Cost ↓ | Utility ↑ |
| Topline | Oracle | 0.91 | 0.38 | 0.83 |
| Naive Baselines | Smallest LLM | 0.59 | 0.2 | 0.55 |
| | Largest LLM | 0.85 | 0.9 | 0.67 |
| | Random | 0.72 | 0.5 | 0.62 |
| Router Baselines | FrugalGPT | 0.71 | 0.34 | 0.64 |
| | FORC | 0.79 | 0.66 | 0.66 |
| | GraphRouter | 0.75 | 0.63 | 0.62 |
| DiSRouter (Ours) | SFT | 0.84 | 0.61 | 0.72 |
| | + RL | 0.84 | 0.53 | **0.73** |

Table 15: **Comparison on a reduced 3-agent system for Cost First scenario.**

| Method | | Cost First: $\alpha = 0.8$ | | |
|---|---|---|---|---|
| | | Accuracy ↑ | Cost ↓ | Utility ↑ |
| Topline | Oracle | 0.91 | 0.38 | 0.61 |
| Naive Baselines | Smallest LLM | 0.59 | 0.2 | 0.43 |
| | Largest LLM | 0.85 | 0.9 | 0.13 |
| | Random | 0.72 | 0.5 | 0.32 |
| Router Baselines | FrugalGPT | 0.61 | 0.21 | 0.44 |
| | FORC | 0.66 | 0.33 | 0.4 |
| | GraphRouter | 0.59 | 0.2 | 0.43 |
| DiSRouter (Ours) | SFT | 0.83 | 0.52 | 0.41 |
| | + RL | 0.75 | 0.29 | **0.52** |

## A.7 EXTERNAL ROUTING CLASSIFIERS

In Section 4.3, we introduced two external binary classifiers for our comparative study: a BERT-based model and an LLM-based model. Here, we detail their training specifics.

**Data** We used the Qwen2.5-7B-Instruct model to generate ground truth labels for classification. The correctness of its greedy-search response on each query across all in-domain datasets was used to create binary labels (1 for "capable," 0 for "not capable"). We randomly sampled 10,000 instances for the training set, the same as those used for the Self-awareness training of Qwen2.5-7B-Instruct. A development set of 500 samples was created. For the final evaluation, we constructed a 10,000-sample test set by randomly sampling 5,000 instances from each class to prevent class bias.

**Training Details**

- **BERT-based Classifier**: We fine-tuned a classifier based on RoBERTa-base (Liu et al., 2019). The training parameters were set as follows: a learning rate of 5e-6, a batch size of 64, and a weight decay of 0.01. The model was trained for two epochs, saving the best checkpoint based on validation performance.
- **LLM-based Classifier**: We fine-tuned a classifier based on LLAMA3-8B-Instruct (Grattafiori et al., 2024). The training parameters were: a learning rate of 1e-5, a batch size of 64, and an evaluation on the development set every 20 steps. The model was also trained for two epochs, saving the best checkpoint.

## A.8 WHETHER TASK CAPABILITY IMPROVED AFTER TRAINING?

Compared with most current routers, our method involves training the LLMs themselves. It would be an unfair comparison if this training procedure were to enhance not only their self-awareness ability but also their inherent task-solving capabilities. To quantitatively assess this, we utilize the **Pass@**$K$ metric (Wang et al., 2022) with $K = 10$. Specifically, for each query, we sampled 10 responses with a temperature $T = 1$. Consistent with the standard definition, a query is considered "answerable by the model" if at least one of the 10 samples is correct. We then measure the performance fluctuation, denoted as $\Delta$**Pass@10**, to quantify whether the model's fundamental problem-solving capability has shifted after training.

Figure 8 plots the $\Delta$Pass@10 for our RL-trained LLMs in the "Balance" scenario. As observed, the performance improvement is sufficiently small to be considered negligible. This aligns with our expectations because no corrective information is introduced during the SFT stage, and the RL stage

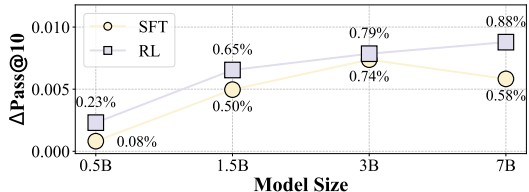

Figure 8: **ΔPass@10 for aligned LLMs with various sizes.** The performance improvement (measured by ΔPass@10) for all models, trained with both SFT and RL, did not exceed 1% and can be considered negligible.

does not bring additional knowledge or expand the models' capability boundary, but only adjusts the models' behavioral preferences as indicated by numerous existing works (Yue et al., 2025; Kim et al., 2025). Therefore, the overall utility increase of our routing system should be attributed to the enhanced self-awareness ability of each model, which is consistent with our motivation.

## A.9 EXPERIMENTS ON INTRINSIC SELF-AWARENESS

The DiSRouter paradigm relies fundamentally on the self-awareness of the constituent agents. In the main text, we utilized our proposed Self-Awareness Training Pipeline to enhance this capability in the Qwen2.5 series models. However, we posit that a truly reliable model should inherently possess a high degree of intrinsic self-awareness, enabling it to autonomously reject queries it cannot correctly solve. In this section, we evaluate and analyze the native self-awareness of LLMs by using prompt engineering to guide models to either answer or reject queries. Our evaluation is two-fold: (1) experiments on open-source Qwen2.5 models, and (2) experiments on SOTA closed-source GPT series LLMs. For both setups, we employ the identical prompt template, as shown in Table 25.

### A.9.1 EXPERIMENTS ON OPEN-SOURCE LLMS

We conducted experiments on four models from the Qwen2.5 series using the seven in-domain datasets described in the main text. To quantify self-awareness, we utilize the following metrics:

- **Answer rate ($\text{Ratio}_{ans}$)**: The average proportion of queries the model chooses to answer.
- **Answer F1**: We treat the decision to answer or reject as a binary classification problem. Using the correctness of the model's response under standard evaluation as the ground truth, we calculate the F1 score. Ideally, a model should choose to answer only those queries it can solve correctly and reject those it cannot.
- **Answered part accuracy ($\text{Acc}_{ans}$)**: The accuracy on the subset of queries the model chose to answer.
- **Rejected part accuracy ($\text{Acc}_{rej}$)**: The accuracy on the subset of queries the model rejected (had it been forced to answer).

An ideal model should exhibit an Answer Rate close to its actual task accuracy, a high Answer F1 score, and an $\text{Acc}_{ans}$ significantly higher than $\text{Acc}_{rej}$. The results are presented in Table 16.

Table 16: **Self-Awareness evaluation of Open-Source LLMs (Qwen2.5-Instruct series).**

| Model Size | $\text{Ratio}_{ans}$ | $\text{F1}_{ans}$ | $\text{Acc}_{ans}$ | $\text{Acc}_{rej}$ |
|---|---|---|---|---|
| 0.5B | 84.77 | 32.7 | 32.07 | 38.07 |
| 1.5B | 6.45 | 23.46 | 60.59 | 58.9 |
| 3B | 39.21 | 48.33 | 70.98 | 69.1 |
| 7B | 33.95 | 41.29 | 75.89 | 73.89 |

As observed, the open-source models exhibit poor intrinsic self-awareness. They demonstrate unstable Answer Rates and low Answer F1 scores, indicating an inability to accurately discern

whether a sample lies within their competence. Furthermore, $Acc_{ans}$ is not significantly superior to $Acc_{rej}$—and in the case of the 0.5B model, it is even lower. This suggests that the rejection behavior of these models is largely random rather than a reflection of true uncertainty. Based on these results, we conclude that current open-source LLMs lack sufficient inherent self-awareness for reliable routing without further alignment.

### A.9.2 EXPERIMENTS ON CLOSED-SOURCE LLMS

Closed-source LLMs, such as the GPT series, are widely regarded as SOTA models. Given their strong task capabilities, we hypothesize they may also possess superior intrinsic self-awareness. We selected three models representing different capability levels and costs: **GPT4.1-NANO**, **GPT4.1-MINI**, and **O4-MINI**. Since these models achieve near-perfect performance on the in-domain tasks used in the main text (rendering them non-discriminative), we conducted this evaluation on a suite of more challenging mathematical reasoning tasks. These include GSM8K (Cobbe et al., 2021), MATH-500 (Hendrycks et al., 2021), AIME 2024/2025 (AI-MO, 2024b), AMC 23 (AI-MO, 2024a), JEEBench (Arora et al., 2023), OlympiadBench (He et al., 2024), and OlympicArena (Huang et al., 2025). For the routing experiments, GSM8K and MATH-500 are treated as in-domain tasks (using their training sets to train router baselines), while the others serve as out-of-domain (OOD) tasks. We assigned normalized costs to these models proportional to their official pricing [1]. The average performance and cost of these models are summarized in Table 17.

Table 17: **Performance and normalized cost configuration for the GPT series models.** ID: in-domain (GSM8K, MATH-500), OOD: out-of-domain (AIME, AMC, etc.).

| Model Name | ID Accuracy | OOD Accuracy | Model Cost |
|---|---|---|---|
| GPT4.1-nano | 86.07 | 42.02 | 0.1 |
| GPT4.1-mini | 89.86 | 61.11 | 0.4 |
| o4-mini | 92.34 | 79.52 | 1.0 |

We employed the same prompting to induce rejection behavior. Figure 9 illustrates the relationship between Answer Rate and Performance for the **GPT4.1-NANO** model, and Figure 10 compares $Acc_{ans}$ against $Acc_{rej}$ across different tasks (**GPT4.1-MINI** exhibited similar trends). Two key observations emerge: (1) There is a strong positive correlation between the LLM's Answer Rate and the task difficulty (performance), implying the model flexibly adjusts its rejection frequency based on difficulty. (2) The accuracy of the answered subset ($Acc_{ans}$) is consistently and significantly higher than that of the rejected subset ($Acc_{rej}$). These findings confirm that these LLMs possess the intrinsic capacity to define their knowledge boundaries—i.e., strong self-awareness.

**DiSRouter with Intrinsic Self-Awareness**  To further validate this, we constructed a *DiSRouter* system using these three models, relying solely on their intrinsic rejection behavior for routing decisions without any additional training. We compared this against three router baselines under the "Balance" scenario ($\alpha = 0.5$). The results are shown in Table 18. Leveraging the inherent self-awareness of the GPT series, *DiSRouter* achieved performance comparable to baselines on in-domain tasks but outperformed them on out-of-domain tasks. This not only validates that an efficient *DiSRouter* system can be constructed without training—provided the underlying LLMs are self-aware—but also validates the poor generalization capability of existing router baselines.

We posit that self-awareness should be viewed as a critical dimension of LLM capability, essential for system reliability. It is plausible that the training of the GPT series already incorporates phases designed to mitigate hallucinations or incorrect responses, a capability that current open-source models appear to lack.

### A.10   EXPERIMENTS ON DIFFERENT MODEL FAMILIES

In the main text, our experiments were conducted exclusively on the Qwen2.5-Instruct series. In this section, we investigate the generalization capabilities of the *DiSRouter* framework and the Self-

---

[1] https://openai.com/api/pricing/

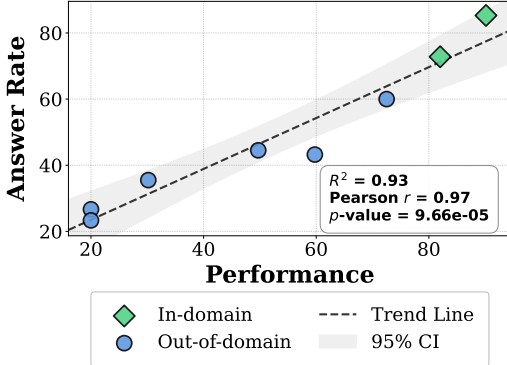
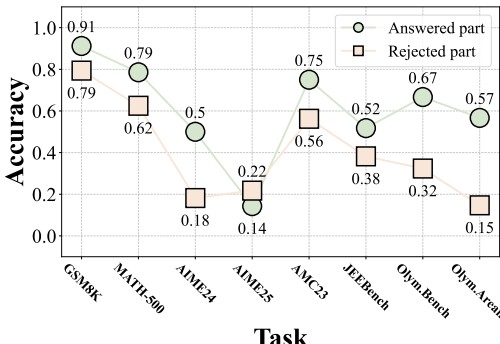

Figure 9: **Correlation between Answer Rate and Model Performance (Task Difficulty) for GPT4.1-nano.** The strong positive correlation indicates that the model autonomously rejects more frequently on harder tasks.

Figure 10: **Comparison of accuracy on answered queries (Acc$_{ans}$) versus rejected queries (Acc$_{rej}$)** The model achieves higher accuracy on questions it chooses to answer, demonstrating effective self-assessment.

Table 18: **Comparison of *DiSRouter* against router baselines on GPT series models under the Balance scenario.** *DiSRouter* demonstrates superior generalization on out-of-domain (OOD) tasks.

| Method | | In Domain | | | Out of Domain | | |
|---|---|---|---|---|---|---|---|
| | | Accuracy ↑ | Cost ↓ | Utility ↑ | Accuracy ↑ | Cost ↓ | Utility ↑ |
| Topline | Oracle | 0.97 | 0.18 | 0.88 | 0.82 | 0.48 | 0.58 |
| Naive Baselines | Smallest LLM | 0.86 | 0.10 | 0.81 | 0.4 | 0.10 | 0.35 |
| | Largest LLM | 0.9 | 1.00 | 0.40 | 0.8 | 1.00 | 0.30 |
| | Random | 0.89 | 0.49 | 0.65 | 0.62 | 0.53 | 0.36 |
| Router Baselines | RouteLLM | 0.86 | 0.10 | **0.81** | 0.42 | 0.10 | 0.37 |
| | FrugalGPT | 0.87 | 0.16 | 0.79 | 0.43 | 0.13 | 0.37 |
| | GraphRouter | 0.92 | 0.4 | 0.72 | 0.57 | 0.39 | 0.38 |
| **DiSRouter** | GPT-series | 0.84 | 0.17 | 0.76 | 0.61 | 0.41 | **0.41** |

Awareness Training pipeline *across* diverse model families. Benefiting from the inherent "plug-and-play" modularity of *DiSRouter*, we modified the original five-agent configuration by removing the 0.5B, 1.5B, and 3B models and substituting them with aligned LLMs from other model families. Specifically, we incorporated Gemma3-1B-Instruct (Team et al., 2025) and Phi4-mini-Instruct (Abdin et al., 2024) to construct a heterogeneous four-agent routing system (comprising models of sizes 1B, 4B, 7B, and 14B).

We applied the Self-Awareness Training pipeline to these two newly introduced models, maintaining the exact training configuration described in the main text to equip them with rejection capabilities. We evaluated the self-awareness of these aligned LLMs using the metrics defined in Appendix A.9.1, with results presented in Table 19. The results demonstrate that our Self-Awareness Training pipeline generalizes effectively across different model families, significantly enhancing model reliability. Meanwhile, the characteristic of scenario adaptability is also generalized, and the behavior of the model is adjusted in line with expectations as the requirements of the scenario change.

Furthermore, we trained router baselines on this four-agent configuration and compared them with *DiSRouter*. The results, summarized in Table 20, indicate that *DiSRouter* consistently achieves superior utility. These findings confirm that the advantages of *DiSRouter* over traditional centralized routers are generalizable, provided the constituent models possess a high degree of self-awareness.

## A.11 EXPERIMENTS ON OPEN-ENDED GENERATIVE TASKS

In the main text, our experiments focused on verifiable tasks where a clear, objective ground truth exists. In this section, we extend the Self-Awareness Training pipeline to open-ended generative tasks, such as text summarization and creative writing. A fundamental premise of any routing system

Table 19: **Self-Awareness evaluation of aligned LLMs from different model families across three scenarios.**

| Backbone Model | Performance First: $\alpha = 0.2$ | | | Balance: $\alpha = 0.5$ | | | Cost First: $\alpha = 0.8$ | | |
|---|---|---|---|---|---|---|---|---|---|
| | $\text{Ratio}_{ans}$ | $\text{Acc}_{ans}$ ↑ | $\text{Acc}_{rej}$ ↓ | $\text{Ratio}_{ans}$ | $\text{Acc}_{ans}$ ↑ | $\text{Acc}_{rej}$ ↓ | $\text{Ratio}_{ans}$ | $\text{Acc}_{ans}$ ↑ | $\text{Acc}_{rej}$ ↓ |
| Gemma-3-1B-instruct | 6.58 | 85.34 | 38.27 | 8.82 | 76.66 | 38.31 | 11.98 | 68.03 | 38.75 |
| Phi-4-mini-instruct | 64.51 | 86.89 | 58.39 | 72.07 | 85.64 | 56.62 | 76.58 | 84.61 | 54.84 |

Table 20: **Performance comparison of *DiSRouter* and baselines on the heterogeneous four-agent system.** *DiSRouter* maintains superior utility, demonstrating strong generalization across model families.

| Method | | Performance First: $\alpha = 0.2$ | | | Balance: $\alpha = 0.5$ | | | Cost First: $\alpha = 0.8$ | | |
|---|---|---|---|---|---|---|---|---|---|---|
| | | Accuracy ↑ | Cost ↓ | Utility ↑ | Accuracy ↑ | Cost ↓ | Utility ↑ | Accuracy ↑ | Cost ↓ | Utility ↑ |
| Topline | Oracle | 0.94 | 0.41 | 0.86 | 0.94 | 0.41 | 0.74 | 0.94 | 0.41 | 0.61 |
| Naive Baselines | Smallest LLM | 0.41 | 0.15 | 0.38 | 0.41 | 0.15 | 0.34 | 0.41 | 0.15 | 0.29 |
| | Largest LLM | 0.85 | 0.90 | 0.67 | 0.85 | 0.90 | 0.40 | 0.85 | 0.90 | 0.13 |
| | Random | 0.71 | 0.56 | 0.60 | 0.71 | 0.56 | 0.43 | 0.70 | 0.56 | 0.25 |
| Router Baselines | RouteLLM | 0.78 | 0.64 | 0.65 | 0.78 | 0.64 | 0.46 | 0.78 | 0.64 | 0.27 |
| | FrugalGPT | 0.81 | 0.68 | 0.67 | 0.66 | 0.35 | 0.49 | 0.49 | 0.19 | 0.34 |
| | GraphRouter | 0.82 | 0.72 | 0.68 | 0.81 | 0.70 | 0.46 | 0.51 | 0.26 | 0.30 |
| DiSRouter (Ours) | SFT | 0.82 | 0.64 | 0.69 | 0.80 | 0.59 | 0.51 | 0.80 | 0.58 | 0.34 |
| | + RL | 0.82 | 0.56 | **0.71** | 0.80 | 0.53 | **0.54** | 0.80 | 0.52 | **0.38** |

is that the task utility must be quantifiable. Unlike verifiable tasks (e.g., QA or Math) where the evaluation score $A(m, x)$ is binary $\{0, 1\}$, open-ended tasks typically yield a continuous quality score $S(m, x) \in [0, 1]$. An unevaluable task renders routing meaningless, as the optimal strategy would trivially default to the cheapest model. Therefore, we generalize the binary scoring function used in the main text to a continuous scoring model, which can be instantiated as a learned reward model or an LLM-as-a-Judge.

### A.11.1 METHODOLOGY ADAPTATION

To accommodate continuous scores, we adapted both the SFT and RL stages of our pipeline.

**SFT Data Construction** In the verifiable setting, we determined rejection based on a fixed probability threshold $\delta_\alpha = 1 - \alpha$. However, continuous scores lack a direct mapping to the preference factor $\alpha$. To address this, we employ a percentile-based thresholding strategy. We define the decision threshold $\delta_\alpha$ as the $(1 - \alpha)$-th percentile of the base model's score distribution on the training set. Which can be formulated as:

$$\delta_\alpha = P_{1-\alpha}(S(m, \mathcal{D}_{train})) \tag{11}$$

For example, in the "Performance First" scenario ($\alpha = 0.2$), the threshold is set to the 80th percentile score ($P_{80}$). Consequently, the model is trained to answer only if its expected output quality belongs to the top 20% of its capability, and to reject otherwise. This data-driven approach effectively preserves the system's Scenario Adaptability within a continuous scoring landscape.

**RL Reward Formulation** We extend the reward function to align with this percentile logic. The reward for a given sample is defined as:

$$r(x, a) = \begin{cases} S(m, x) & \text{if answer} \\ P_{1-\alpha} & \text{if reject} \end{cases} \tag{12}$$

Here, $P_{1-\alpha}$ serves as the baseline reward for rejection. The model learns to answer only when its generated response quality $S(m, x)$ is expected to exceed the threshold $P_{1-\alpha}$, mirroring the logic of the verifiable task setting.

### A.11.2 EXPERIMENTAL SETUP AND RESULTS

We validated the effectiveness of this extended pipeline on the text summarization task, utilizing the SAMSum (Gliwa et al., 2019) and XSum (Narayan et al., 2018) datasets. We employed BERTScore (Zhang et al., 2020) as the evaluation metric, using the recommended DEBERTA-XLARGE-MNLI (He et al., 2021) as the backbone. The baseline BERTScore performance of the Qwen2.5-Instruct series is presented in Table 21. We observe that performance is relatively saturated across model sizes—a common challenge in evaluating open-ended generation. While more sophisticated reward models could offer finer granularity, our primary objective is to assess whether Self-Awareness Training can effectively teach the model to reject samples prone to lower scores based on the given metric signal.

Table 21: **Baseline performance (BERTScore) of Qwen2.5-Instruct series models on SAMSum and XSum datasets.** The relatively narrow performance gap across model sizes reflects the characteristics of the metric and the task.

| Task | 0.5B | 1.5B | 3B | 7B |
|---|---|---|---|---|
| SAMSsum | 67.5 | 68.97 | 69.64 | 70.71 |
| XSum | 64.4 | 66.21 | 65.04 | 67.34 |
| Average | 65.95 | 67.59 | 67.34 | 69.03 |

We conducted Self-Awareness Training on four model sizes, maintaining the same hyperparameters as in the main text. The training dataset consisted of 4,000 samples per scenario (Total = 12,000 samples). The Answer Rate and the BERTScore comparison between answered and rejected subsets are illustrated in Figure 11 and Figure 12, respectively.

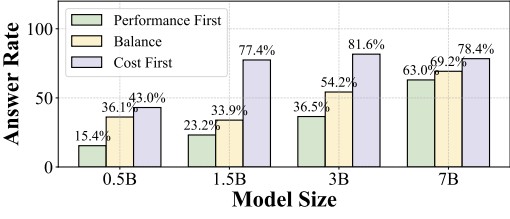

Figure 11: **Answer Rate of models on summarization tasks across different scenarios.** The models exhibit consistent scenario adaptability, answering fewer queries as the requirement for performance (high scores) increases.

Figure 12: **Comparison of BERTScore between answered and rejected queries.** The models achieve consistently higher scores on queries they choose to answer.

The results indicate that: (1) The trained models retain robust Scenario Adaptability, flexibly adjusting their answer rates according to the preference factor $\alpha$; and (2) The average BERTScore of the answered subset is significantly higher than that of the rejected subset. This demonstrates that the model successfully learns to anticipate its performance and reject lower-quality outputs. These findings suggest that our Self-Awareness Training pipeline generalizes effectively to open-ended tasks, provided a reliable reward model is available.

## A.12 PROMPTS AND TEMPLATES

### A.12.1 SCENARIO PROMPTS

To imbue each *DiSRouter*-aligned agent with Scenario Adaptability, we prepend scenario-specific instructions to the prompt of each query, as detailed in Table 22. This ensures that the agent's behavior aligns with the requirements of the given scenario.

Table 22: **Prompts for each scenario.**

| Scenario | Prompt |
|---|---|
| Performance First | Scenario: Performance First. You must ensure your answer is correct. Only choose to answer if you are completely certain of your ability to answer the question; otherwise, choose not to answer. |
| Balance | Scenario: Balance. You need to strike a balance between answering and not answering. You can choose to answer questions you are confident in and choose not to answer those you are uncertain about. |
| Cost First | Scenario: Cost First. You need to answer questions as much as possible. Choose to answer as long as you believe you have the ability to answer correctly; only choose not to answer if you feel there is absolutely no possibility of giving a correct answer. |

### A.12.2 SELF-AWARENESS TRAINING RESPONSE TEMPLATES

When conducting Self-Awareness Training, a model can choose to either answer or reject a query. In the SFT stage, we use distinct response templates for these two behaviors, as shown in Table 23. For rejection, the model directly responds with "I don't know!" For answering, we use an extraction LLM (Qwen2.5-14B-Instruct) to extract the thinking steps and the final answer from the original Qwen model's response on the training samples, and then use the "Answer" template in Table 23 to construct the training data. The prompt used for the extraction of the large model is shown in Table 24.

Table 23: **Response template for training data.**

| Action | Template |
|---|---|
| Answer | Let's think step by step: <**thinking steps**>. The final answer is: <**final answer**>. |
| Reject | I don't know! |

Table 24: **Instruction for the Extraction LLM.** The model extracts thinking steps and final answer from the responses of the original LLM.

| Extraction Prompt |
|---|
| There is a question and the corresponding output from a model. I would now like you to help me extract the reasoning part and the final answer from the model's output. Please note that only the reasoning part and the final answer is needed, and any other content should be excluded, and do not generate any other prefix or suffix. Use the following template to reply: The Extracted Reasoning Parts: <>. The Extracted Final Answer: <>. Now, start the task: Input Question:\n<question>.\n Model Output:<output> . The content you extracted: |

### A.12.3 REJECTION PROMPT FOR INTRINSIC SELF-AWARENESS EVALUATION

Table 25: **The prompt template used to evaluate intrinsic self-awareness.** The same prompt is applied to both Open-source and Closed-source experiments.

| Rejection Prompt |
|---|

You are a "Hyper-Cautious Expert" AI. Your primary directive is to answer only when you are 100% certain of the entire, complete, and accurate answer. You must follow these rules:

1. If you have **ANY DOUBT** at all—because the question is ambiguous, too complex, unverifiable, an unsolved problem, or you are not 100% sure of the correct steps—you **MUST** refuse. Your **ONLY** response must be the three-word phrase: "I don't know!"

2. If you are **100% CERTAIN** you can provide a flawless, correct, and complete solution, you **MUST** provide the answer. Your answer **must** follow this exact format:

* It must begin with "Let's think step by step."
* Your reasoning must be **concise and accurate**, focusing **only on the key logical steps**.
* Avoid verbose explanations. Your entire response must be as brief as possible, ideally under 2000 tokens.
* It must conclude with the final answer enclosed in "\\boxed{}". All mathematical formulas and symbols you output should be represented with LaTeX!

**[CRITICAL INSTRUCTIONS]**
* **DO NOT** apologize if you refuse.
* **DO NOT** explain *why* you don't know.
* **DO NOT** provide partial answers.
* Your choice is binary: either a concise, reasoned, boxed answer (Rule 2) or a simple refusal (Rule 1).

**[BEHAVIORAL EXAMPLES]**
··· [FIVE SHOTS]

**[TASK BEGINS]**
Now, apply these rules strictly to the following question.

