# OpenReview forum: "DiSRouter: Distributed Self-Routing for LLM Selections"
_ICLR.cc/2026/Conference — ICLR 2026 Poster_

### Official Review · Reviewer_JRcd · 2025-10-28

**Soundness:** 3
**Presentation:** 3
**Contribution:** 3
**Rating:** 8
**Confidence:** 3

**Summary:**

A recent strategy to minimize compute involves routing user prompts to an LLM which has just the right size to handle the request, instead of always prompting the most capable model.

Currently, the most popular routing strategy involves using a routing model, which determines which LLM to prompt with the user's request. This paper proposes to instead integrate the routing process directly into the available LLMs. Specifically, the LLMs are trained to defer to the next biggest model when they deem they are not capable of providing a correct answer. The paper shows that this results a better balance between cost and accuracy.

**Strengths:**

- The problem formulation is clear and the motivation is sound.
- This idea will be of interest to the community.
- The results show that the idea has merit and should be further investigated.

Overall, I believe that this paper should be accepted, although authors should address the weaknesses mentioned below.

**Weaknesses:**

My main issue is that the authors barely discuss/explore the (potential) limitations of the method. For instance, the method first prompts the smallest model. Because of its size, it might sometimes struggle to properly reason whether it is capable of accurately answering a given question. Although the results seem positive, have the authors observed such phenomena, our found that for certain type of tasks this happened at a greater rate?

Another issue is the computational cost of the method. The most difficult prompts must go through all models before the final largest model answers. This is the worse case scenario, and it is a downside of the method. Authors do not talk about this much. The method does seem more costly than certain router baselines, but performs slightly better. Have the authors considered potentially skipping an LLM when say the first model's confidence is very low, or outputting more nuanced reject answers, such as "I don't now.", "I really don't know.", and depending on such output, skip an LLM or two in the chain?

Another important question is how does internalizing the routing decision degrade a model's capabilities compared to standard instruction-tuning. It is pretty important to know if this creates a gap and if so, how big is it.

Aside from discussing the (potential) limitations of the method, the paper's methodology section would also gain from being clearer. The current description is somewhat high level and leaves a lot of the details unspecified. For instance, during RL fine-tuning, how are the model's answers judged as correct or incorrect?

Finally, it would be interesting to know how the authors designed the prompts for the "Performance First", "Balance", and "Cost First" scenarios in Table 7 of the Appendix. Were multiple prompts tested or did the authors simply use the first prompt that came to mind?

Addressing these aspects would help strengthen the paper.

**Questions:**

None

---

> ### Author Response · Authors · 2025-11-26
>
> ## **W1: The method first prompts the smallest model. Because of its size, it might sometimes struggle to properly reason whether it is capable of accurately answering a given question. Although the results seem positive, have the authors observed such phenomena, our found that for certain type of tasks this happened at a greater rate?**
>
> We appreciate this insightful question. We explicitly investigated whether the smallest model (0.5B), given its limited capacity, struggles to reason about its own capabilities. Our findings reveal that while small base models indeed exhibit "fuzzy" knowledge boundaries, our **Self-Awareness Training significantly "sharpens" these boundaries**, enabling effective discrimination consistent across varying task types.
>
> ### **1. Inherent Ambiguity in Base Models**
>
> First, we quantified the "intrinsic knowledge boundary" of the base models. We performed 10-shot stochastic sampling for each query to calculate the **Sampling Accuracy** (frequency of correct answers).
>
> - **Metric:** Accuracy $\approx 0$ or $1$ indicates a clear boundary (definitely incorrect or correct). Accuracy $\approx 0.5$ indicates high uncertainty.
> - **Observation (Table R1):** As suspected, the **0.5B Base Model** exhibits a high degree of ambiguity, with a significant proportion of samples falling in the fuzzy $[0.2, 0.7]$ range. In contrast, the **7B Base Model** shows a **clear "U-shaped" distribution**, with samples clustered at the extremes.
>
> *Table R1: Distribution of 10-shot Sampling Accuracy for Base Models (0.5B vs. 7B).
> (The header represents the sampling accuracy bin, and values represent the proportion of samples in that bin.)*
>
> | Size |  0 | 0.1 | 0.2 | 0.3 | 0.4 | 0.5 | 0.6 | 0.7 | 0.8 | 0.9 | 1 |
> | --- | --- | --- | --- | --- | --- | --- | --- | --- | --- | --- | --- |
> | 0.5B | 0.06 | 0.12 | 0.15 | 0.15 | 0.14 | 0.12 | 0.1 | 0.07 | 0.05 | 0.02 | 0.01 |
> | 7B | 0.07 | 0.03 | 0.02 | 0.02 | 0.02 | 0.02 | 0.02 | 0.03 | 0.04 | 0.08 | 0.65 |
>
> ### **2. Sharpening Boundaries via Training**
>
> To analyze the impact of our training, we evaluated the **Aligned 0.5B Agent**. We forced it to answer all queries (via prefix injection) and analyzed the sampling accuracy distribution separately for the subsets it chose to Answer versus Reject. The results in **Table R2** are striking:
>
> - **Strong Discrimination:** The distributions are highly polarized. The "Answered" subset is heavily concentrated in the high-accuracy region ($>0.8$), while the "Rejected" subset is concentrated in the low-accuracy region ($<0.3$).
> - **Boundary Sharpening:** Even for the smallest model, the training effectively suppresses the "fuzzy middle ground," allowing the model to conservatively reject uncertain queries while confidently answering solvable ones.
>
> *Table R2: Distribution of Sampling Accuracy for the Aligned 0.5B Agent, partitioned by its decision (Answered vs. Rejected).*
>
> | Split | 0 | 0.1 | 0.2 | 0.3 | 0.4 | 0.5 | 0.6 | 0.7 | 0.8 | 0.9 | 1 |
> | --- | --- | --- | --- | --- | --- | --- | --- | --- | --- | --- | --- |
> | Rejected | 0.17 | 0.15 | 0.13 | 0.1 | 0.09 | 0.07 | 0.06 | 0.06 | 0.06 | 0.05 | 0.06 |
> | Answered | 0.14 | 0.04 | 0.01 | 0.01 | 0.01 | 0.01 | 0.01 | 0.03 | 0.05 | 0.14 | 0.56 |
>
> ### **3. Consistency Across Task Types**
>
> Regarding task variation, we further broke down the performance of the 0.5B model across different in-domain task categories under the *Balance* scenario (**Table R3**).
>
> - **Consistent Gap:** The model maintains consistent self-assessment performance across diverse tasks. In all categories, the accuracy of answered queries ($Acc_{ans}$) is significantly higher than the potential accuracy of rejected queries ($Acc_{rej}$, obtained via forced answering).
> - **Conclusion:** This confirms that the model's self-awareness is not limited to specific easy tasks but is a generalized capability that functions reliably even on this compact architecture.
>
> *Table R3: Task-specific Self-Assessment Performance of the 0.5B Agent (Balance Scenario).
> ($Acc_{ans}$: Accuracy on queries the model chose to answer; $Acc_{rej}$: Accuracy on queries the model rejected but was forced to answer.)*
>
> | Task | Answer rate | Acc$_{ans}$ | Acc$_{rej}$ |
> | --- | --- | --- | --- |
> | gsm8k | 10.69 | 87.94 | 37.86 |
> | ai2_arc_challenge | 9.81 | 72.17 | 35.1 |
> | mmlu | 6.08 | 83.61 | 33.29 |
> | drop | 4.29 | 76.28 | 27.9 |
> | cosmosqa | 9.08 | 78.23 | 32.76 |
> | openbookqa | 11.2 | 82.14 | 39.41 |
> | race_high | 12.44 | 90.11 | 40.55 |

---

> > ### Author Response · Authors · 2025-11-26
> >
> > ## **W2: Another issue is the computational cost of the method. The method does seem more costly than certain router baselines, but performs slightly better. Have the authors considered potentially skipping an LLM when say the first model's confidence is very low, or outputting more nuanced reject answers, such as "I don't now.", "I really don't know.", and depending on such output, skip an LLM or two in the chain?**
> >
> > We appreciate this practical perspective, which was echoed by other reviewers. To address concerns regarding latency and overhead, we have added a detailed statistical analysis of actual inference and routing costs in **Appendix A.1** of the revised paper.
> >
> > ### **Latency Analysis**
> >
> > 1. **Negligible Routing Overhead:** Our empirical data (**Table 6**) shows that although a query may be rejected by multiple agents before being solved, the **cumulative routing cost** (time spent generating rejections) remains negligible. Even in the worst-case scenario (routing to the final agent), this overhead constitutes less than **5%** of the actual inference cost of generating the final answer (**Table 7**).
> > 2. **Competitive Latency:** We also benchmarked **System Latency** (Time to First Token, **Table R4 and Table 8 in paper**). Results indicate that while DiSRouter incurs slightly higher latency than predictive routers (e.g., RouteLLM) which use lightweight classifiers, it is significantly faster than generative baselines that rely on full model outputs or verification steps (e.g., FrugalGPT, AutoMix).
> >
> > *Table R4: Time to First Token (s) Results Comparison for Different Routing Systems.*
> >
> > |  | RouterLLM | FrugalGPT | Automix | FORC | GraphRouter | DiSRouter |
> > | --- | --- | --- | --- | --- | --- | --- |
> > | Average | 0.048 | 2.620 | 3.475 | 0.070 | 0.024 | 0.081 |
> > | Worst Case | 0.063 | 9.565 | 11.34 | 0.076 | 0.028 | 0.241 |
> >
> > ### **Design of Cross-Level Routing**
> >
> > We appreciate this forward-looking suggestion. We have actually initiated preliminary research into it, where agents are conditioned on system-level information to potentially route difficult queries directly to a much larger model, skipping intermediate steps. However, our initial experimental results showed no significant improvement in **system Utility**. We attribute this to two primary factors:
> >
> > 1. **Complexity of Routing Precision:** Accurately determining which specific future agent (e.g., skipping from 1.5B to 14B vs. 7B) is required is a significantly harder problem than the binary decision of determining one's own incompetence. Our current algorithm for this fine-grained skipping is not yet precise enough to outweigh the risks of over-routing (using too large a model) or under-routing.
> > 2. **Marginal Cost Benefits:** As discussed in our response regarding routing overhead (and in **Appendix A.1**), the cost of traversing the cascade is already very low (<5% of inference cost). Therefore, the latency savings gained by "skipping a level" are minimal. The primary theoretical value of skipping would be to bypass intermediate agents that might confidently answer incorrectly. However, since our Self-Awareness Training already ensures agents reject when uncertain, this "error skipping" value is also diminished.
> >
> > We consider cross-level routing a promising direction for optimizing extremely deep cascades or heterogeneous agent pools and plan to explore it further in future work.

---

> > > ### Author Response · Authors · 2025-11-26
> > >
> > > ## **W3: Another important question is how does internalizing the routing decision degrade a model's capabilities compared to standard instruction-tuning. It is pretty important to know if this creates a gap and if so, how big is it.**
> > >
> > > We thank the reviewer for raising this crucial question. It is vital to distinguish between two orthogonal attributes: **Task Capability** (the intrinsic knowledge to solve a specific problem) and **Self-Awareness** (the meta-cognitive ability to recognize knowledge boundaries).
> > >
> > > ### **1. Conceptual Distinction**
> > >
> > > Our Self-Awareness Training Pipeline is explicitly designed to **enhance Self-Awareness, not Task Capability**. To ensure a fair comparison of routing mechanics and strictly evaluate the "routing logic," we deliberately avoided introducing additional supervision signals for task enhancement.
> > >
> > > - **SFT Stage:** We only trained on samples the model could *already* solve correctly (filtering out new knowledge).
> > > - **RL Stage:** Responses were sampled from the model's own policy distribution.
> > > As discussed in **Appendix A.8**, this ensures that any performance gain in the system comes from *better routing*, not *better base models*.
> > >
> > > ### **2. Comparative Experiment: Standard SFT vs. Self-Awareness**
> > >
> > > To rigorously quantify if internalizing the routing decision degrades capabilities compared to standard instruction tuning, we conducted a comparative experiment using the **3B model**.
> > >
> > > - **Baselines:**
> > >     1. **Base Model:** The original pre-trained checkpoint.
> > >     2. **Standard SFT:** The model fine-tuned on the same dataset but with standard cross-entropy loss on ground-truth answers (optimizing for correctness).
> > >     3. **DiSRouter (Aligned):** Our model trained via Self-Awareness SFT + RL.
> > > - **Evaluation Method:** To measure the latent capability of the DiSRouter model, we **forced it to generate an answer for every query** (by injecting answer prefixes), effectively suppressing its rejection mechanism.
> > >
> > > The results are presented in **Table R5**.
> > >
> > > *Table R5: Capability Comparison on the 3B Model: Standard SFT vs. DiSRouter (Forced Answering). "Full Set" indicates global accuracy. For DiSRouter, we further partition the Full Set based on the model's original decision to Answered or Rejected, to visualize capability distribution.*
> > >
> > > | Index | Task | Split | gsm8k | ai2_arc_challenge | mmlu | drop | cosmosqa | openbookqa | race_high | Average |
> > > | --- | --- | --- | --- | --- | --- | --- | --- | --- | --- | --- |
> > > | 1 | Base LLM | Full | 0.84 | 0.76 | 0.64 | 0.63 | 0.67 | 0.81 | 0.78 | 0.73 |
> > > | 2 | Standard SFT | Full | 0.84 | 0.83 | 0.67 | 0.71 | 0.71 | 0.85 | 0.82 | 0.77 |
> > > | 3 | Self-Awareness Training | Full | 0.83 | 0.77 | 0.64 | 0.59 | 0.68 | 0.79 | 0.77 | 0.72 |
> > > | 4 | Self-Awareness Training | Answered | 0.91 | 0.91 | 0.87 | 0.8 | 0.91 | 0.94 | 0.93 | 0.89 |
> > > | 5 | Self-Awareness Training | Rejected | 0.54 | 0.54 | 0.42 | 0.48 | 0.57 | 0.6 | 0.58 | 0.53 |
> > >
> > > ### **3. Analysis of Results**
> > >
> > > The comparison yields two key insights:
> > >
> > > 1. **Global Accuracy (Capability Maintenance):** Standard SFT significantly improves accuracy on the Full Set (comparing Row 2 vs. 1). In contrast, our Self-Awareness trained model shows **global accuracy comparable to the Base model** (comparing Row 3 vs. 1). This confirms that our training does not inject new task knowledge, nor does it significantly degrade the model's latent knowledge; it simply alters the *expression* of that knowledge.
> > > 2. **Performance Separation (Reliability):** However, DiSRouter demonstrates superior **performance separation**.
> > >     - **Answered Subset:** On the queries it originally chose to answer, DiSRouter achieves exceptional accuracy, **outperforming even the Standard SFT model**.
> > >     - **Rejected Subset:** Conversely, on the queries it chose to reject, its latent accuracy is extremely low.
> > >
> > > **Conclusion:** The training does not "degrade" the model's intelligence; rather, it effectively concentrates the model's active competence into a high-precision zone. In practical applications, we view these approaches as complementary: one could combine Standard SFT (to boost capability) with Self-Awareness Training (to ensure reliability) to achieve an optimal agent.

---

> > > > ### Author Response · Authors · 2025-11-26
> > > >
> > > > ## **W4: Aside from discussing the (potential) limitations of the method, the paper's methodology section would also gain from being clearer. The current description is somewhat high level and leaves a lot of the details unspecified. For instance, during RL fine-tuning, how are the model's answers judged as correct or incorrect?**
> > > >
> > > > We appreciate the reviewer's constructive suggestion regarding the clarity of our methodology. We agree that specifying the reward definition is essential for reproducibility, and we will refine **Section 3** in the final revision to provide these details explicitly.
> > > >
> > > > ### **To clarify the reward mechanism**
> > > >
> > > > **1. Verifiable Tasks (Primary Experiments)**
> > > >
> > > > In our main experiments (Mathematical Reasoning and Multiple-Choice QA), we focused on **verifiable tasks** where objective ground truths exist.
> > > >
> > > > - **Mechanism:** We utilized standard rule-based evaluation scripts (e.g., regex matching for math answers or option extraction for multiple-choice).
> > > > - **Reward Signal:** When the agents choose to answer, the environment executes these scripts to assign a **binary reward**: $r=1$ if the model's output matches the ground truth, and $r=0$ otherwise. This binary feedback loop is standard in reasoning and routing literature as it provides a clear optimization target.
> > > >
> > > > **2. Open-Ended Tasks (Extension)**
> > > > However, motivated by suggestions from other reviewers, we have extended the framework to **open-ended tasks** (e.g., Summarization) where correctness is not binary.
> > > >
> > > > - **Mechanism:** In these cases, we employ a reference-based metric (e.g., **BERTScore**) to serve as the reward model.
> > > > - **Reward Signal:** The reward becomes a **continuous value** (normalized to $[0, 1]$). As detailed in **Appendix A.11**, our results confirm that the RL training effectively optimizes the model's self-awareness using this continuous signal, demonstrating the **DiSRouter's generalizability beyond rigid verifiable tasks.**
> > > >
> > > >
> > > > ## **W5: Finally, it would be interesting to know how the authors designed the prompts for the "Performance First", "Balance", and "Cost First" scenarios in Table 7 of the Appendix. Were multiple prompts tested or did the authors simply use the first prompt that came to mind?**
> > > >
> > > > Regarding the scenario prompts, we adopted a **minimalist design strategy**. We did not conduct extensive prompt engineering or search for "optimal" phrasing. Instead, we aimed for concise, natural language instructions that intuitively convey the distinct preferences of each scenario (e.g., emphasizing "correctness" for Performance First vs. "answering as much as possible" for Cost First).
> > > >
> > > > Our hypothesis—which our results confirm—is that the **specific wording is secondary**. Because the Self-Awareness Training pipeline explicitly aligns the model's behavior (via conditioned rewards) to these specific prompt tokens, the model learns to associate the semantic meaning of the prompt with the desired rejection threshold during training. Therefore, the system is robust to the prompt choice, provided the instruction is consistent during training and inference.

---

### Official Review · Reviewer_dfTT · 2025-10-29

**Soundness:** 2
**Presentation:** 3
**Contribution:** 2
**Rating:** 4
**Confidence:** 3

**Summary:**

The paper introduces a distributed routing paradigm, enabled by a concrete self-awareness training pipeline, to address LLM cost-performance trade-offs. However, the core claims are undermined by significant methodological flaws, including the failure to account for cumulative latency and a lack of generalization evidence beyond a single, homogeneous model family for narrow, verifiable tasks.

**Strengths:**

1. The paper introduces a "Distributed Self-Routing" framework, replacing the external centralized router with intrinsic "self-awareness."

2. It provides a concrete technical contribution with the two-stage "Self-Awareness Training" pipeline (SFT + RL), enabling models to internally assess their knowledge boundaries and execute a "reject" action.

3. The framework is inherently modular ("plug-and-play").

**Weaknesses:**

1. The core utility metric (Utility = Performance - α * Cost) is flawed. It completely ignores the significant cumulative latency incurred by the cascade structure. A query rejected multiple times will have a real-world response time far exceeding that of a single-shot centralized router, a critical cost factor this paper overlooks.

2. The experiments are confined to a homogeneous model pool (all Qwen-family models). This is a major threat to validity. The paper provides no evidence that the "self-awareness" training paradigm generalizes to a realistic, heterogeneous ecosystem of diverse model architectures (e.g., Llama, Mistral, Gemma), which is the primary motivation for such a system.

3. The entire methodology (self-awareness training, binary accuracy evaluation) is fundamentally dependent on tasks with clear, verifiable answers (e.g., math, multiple-choice QA). It is unclear how this approach would apply to open-ended generative tasks (e.g., summarization, creative writing) where "correctness" is ill-defined, severely limiting its practical scope.

4. The evidence supporting the superiority of "intrinsic assessment" is marginal. In Table 5, the purpose-trained 7B DiSRouter (0.80 F1) barely outperforms a general-purpose 8B external classifier (0.77 F1). This negligible gap fails to provide a strong justification for the added complexity of the distributed training approach over a simpler, well-trained external router.

**Questions:**

None

---

> ### Author Response · Authors · 2025-11-26
>
> ## **W1: The core utility metric (Utility = Performance - α * Cost) is flawed. It completely ignores the significant cumulative latency incurred by the cascade structure. A query rejected multiple times will have a real-world response time far exceeding that of a single-shot centralized router, a critical cost factor this paper overlooks.**
>
> We appreciate the reviewer’s rigorous scrutiny regarding the formulation of our Utility metric. We wish to clarify that the omission of routing overhead in the original formula was a **deliberate simplification for mathematical conciseness**, predicated on our analysis that this term is **statistically negligible** compared to inference costs, as we’ve declared in **Section 2.1** and discussed in **Appendix A.1**.
>
> While the omission was a statistically justified simplification, we agree with the reviewer that explicitly formalizing this term enhances theoretical rigor. We will **revise the formal definition** of the cost function in the later version of manuscript to explicitly include the routing overhead term, ensuring the methodology is mathematically flawless.
>
> To validate this design choice and further address your concern, we provide a detailed breakdown proving that the "cumulative latency" does not materially affect the cost-benefit landscape.
>
> ### **1. Empirical Analysis of Routing Overhead**
>
> As detailed in **Appendix A.1**, we conducted a comprehensive statistical analysis of inference and routing costs.
>
> - **Marginal Cost Ratio:** Our empirical data confirms that the "routing cost" (time spent on rejections) is minimal compared to the "inference cost" (time spent generating the solution) (**Table 7**). Even in the worst-case path (traversing the entire chain to the final agent), the cumulative time spent on rejections constitutes **less than 5%** of the time required for the final answer generation. This is structurally inherent: a rejection is a short sequence (e.g., "I don't know!"), whereas a valid answer is typically a long CoT sequence.
> - **Latency Analysis:** We further benchmarked System Latency (Time to First Token) in **Table R1**. While DiSRouter is slightly slower than lightweight predictive routers (e.g., RouteLLM), it is **significantly faster** than generative verification baselines (e.g., FrugalGPT, AutoMix), confirming its practical efficiency.
>
>     *Table R1: Comparison of Time to First Token (s) between routing systems.*
>
>     |  | RouterLLM | FrugalGPT | Automix | FORC | GraphRouter | DiSRouter |
>     | --- | --- | --- | --- | --- | --- | --- |
>     | Average | 0.048 | 2.620 | 3.475 | 0.070 | 0.024 | 0.081 |
>     | Worst Case | 0.063 | 9.565 | 11.34 | 0.076 | 0.028 | 0.241 |
>
> ### **2. Sensitivity Analysis: Impact on System Utility**
>
> To directly address the reviewer’s concern, we re-evaluated our system using an Adjusted Utility Metric that explicitly subtracts the routing overhead cost:
>
> $$Utility_{adjusted} = Performance - \alpha \cdot (Cost_{inference} + Cost_{routing})$$
>
> The comparison between the original and adjusted metrics is presented in **Table R2**.
>
> Table R2: Sensitivity Analysis of System Utility: A comparison between original metrics (inference cost only) and adjusted metrics (incorporating cumulative routing overhead, **marked with a \***).
>
> | Scenario | Alpha | Accuracy | Cost | Cost* | Utility | Utility* |
> | --- | --- | --- | --- | --- | --- | --- |
> | Performance First | 0.2 | 0.81 | 0.396 | 0.413 | 0.731 | 0.727 |
> | Balance | 0.5 | 0.77 | 0.321 | 0.333 | 0.610 | 0.604 |
> | Cost First | 0.8 | 0.75 | 0.289 | 0.299 | 0.519 | 0.511 |
> - **Minimal Deviation:** The results demonstrate that including the routing cost has a minimal impact on the overall System Utility. Even in the sensitive "Cost First" scenario, the overall utility drops by only approximately **1.5%**.
> - **Conclusion:** This confirms that the cumulative latency, while physically present, does not significantly alter the cost-benefit landscape or invalidate the comparative advantages of DiSRouter.

---

> > ### Author Response · Authors · 2025-11-26
> >
> > ## **W2: The experiments are confined to a homogeneous model pool (all Qwen-family models). This is a major threat to validity. The paper provides no evidence that the "self-awareness" training paradigm generalizes to a realistic, heterogeneous ecosystem of diverse model architectures (e.g., Llama, Mistral, Gemma), which is the primary motivation for such a system.**
> >
> > We are grateful for this insightful comment concerning the **cross-architecture applicability** of DiSRouter. Our new experiments confirm the strong generalization capability of both the Self-Awareness Training Pipeline and the overall DiSRouter architecture.
> >
> > ### **Experimental Validation of Cross-Architecture Generalization**
> >
> > To rigorously validate this, we conducted a new set of experiments that integrated models from distinct families into our system.
> >
> > 1. **Agent Integration:** We applied our Self-Awareness Training to two new models, **Gemma3-1B-Instruct** (1B) and **Phi4-mini-Instruct** (4B), to enhance their rejection capabilities.
> > 2. **System Modification:** Leveraging the **"plug-and-play" modularity** of DiSRouter, we modified our original five-agent routing system (0.5B-1.5B-3B-7B-14B) by removing the three smallest models and substituting them with these two new models. This resulted in a **heterogeneous four-agent routing system (1B-4B-7B-14B)**, which was then benchmarked against other router baselines.
> >
> > The results decisively confirm the generalizability across different model families:
> >
> > 1. **Effective Adaptation:** Post-training, both the Gemma3 and Phi4 models demonstrated **strong self-assessment capabilities** and **Scenario Adaptability**, successfully distinguishing in-competence queries and adhering to cost constraints. (**Table R3**.)
> >
> >     *Table R3: A summary of model self-awareness ability in the Balance scenario. $Ratio_{ans}$ refer to the answer rate, $Acc_{ans}$ and $Acc_{rej}$ refer to the accuracy on the subset of queries the model chose to answer and reject, respectively.*
> >
> >     | Backbone Model | Ratio$_{ans}$ | Acc$_{ans}$ | Acc$_{rej}$ |
> >     | --- | --- | --- | --- |
> >     | Gemma-3-1B-instruct | 8.82 | 76.66 | 38.31 |
> >     | Phi-4-mini-instruct | 72.07 | 85.64 | 56.62 |
> > 2. **Superior Utility:** The resulting heterogeneous DiSRouter system continued to achieve **significantly superior routing Utility**, delivering higher performance at a lower computational cost compared to centralized router baselines. (**Table R4**)
> >
> >     *Table R4: Utility of routing systems.*
> >
> >     | Method | Accuracy | Cost | Utility |
> >     | --- | --- | --- | --- |
> >     | Oracle | 0.94 | 0.41 | 0.74 |
> >     | Random | 0.71 | 0.56 | 0.43 |
> >     | GraphRouter | 0.81 | 0.7 | 0.46 |
> >     | DiSRouter | 0.80 | 0.53 | 0.54 |
> >
> >     Full details could be found in **Tables 19 and 20** of the revised paper.
> >
> >
> > ### **Validation of Distributed Design**
> >
> > Crucially, this experiment further underscores the key advantage of DiSRouter’s distributed design: modifying the agent pool only required the **one-time training** of the new agents (Gemma3 and Phi4). The existing agents (Qwen2.5-Instruct 7B/14B) were **reused directly without any retraining**, validating the system's long-term maintainability and scalability.
> >
> > We have included the complete experimental setup and detailed results in **Appendix A.10 of the revised manuscript** and encourage the reviewer to examine them.

---

> ### Author Response · Authors · 2025-11-26
>
> ## **W3: The entire methodology (self-awareness training, binary accuracy evaluation) is fundamentally dependent on tasks with clear, verifiable answers (e.g., math, multiple-choice QA). It is unclear how this approach would apply to open-ended generative tasks (e.g., summarization, creative writing) where "correctness" is ill-defined, severely limiting its practical scope.**
>
> We thank the reviewer for highlighting the importance of **open-ended tasks**, which are indeed critical for evaluating broader generalization. We have extended our framework to address such tasks and conducted new generalization experiments on **text summarization**. The results confirm that our **Self-Awareness Training Pipeline generalizes effectively**, enabling models to distinguish high-quality generation from low-quality generation even when the metric is continuous rather than binary.
>
> We have included the full experimental setup and detailed results in **Appendix A.11** of the revised paper.
>
> ### **1. Methodological Adaptation: From Binary to Continuous Scoring**
>
> As detailed in **Appendix A.11.1**, we adapted our pipeline to handle the continuous nature of open-ended tasks (where quality is not a simple {0, 1}):
>
> **Percentile-based SFT Threshold:** Unlike verifiable tasks with binary outcomes, summarization quality (e.g., BERTScore) is continuous. We replaced the fixed probability threshold with a **percentile-based threshold**. Specifically, the rejection threshold $\delta$ is dynamically set to the $(1-\alpha)\text{-th}$ percentile of the model's score distribution on the training set.
>
> **Continuous RL Reward:** Similarly, the reward for answering is defined as the continuous score output by the evaluation metric. The reward for rejecting is anchored to the $(1-\alpha)\text{-th}$ percentile baseline. This effectively normalizes the reward signal, allowing the model to learn relative quality distinctions.
>
> ### **2. Experimental Validation on Summarization**
>
> We evaluated this approach on two standard datasets, **SAMSum** and **XSum**, using **BERTScore** as the quality metric (**Appendix A.11.2**).
>
> - **Strong Discriminative Capability:** Post-alignment, the models demonstrated significant self-awareness. The average BERTScore of queries the model chose to Answer was significantly higher than that of the queries it Rejected (see **Table R5 and Figure 12**), proving it can successfully identify its own high-quality outputs.
>
>     *Table R5: BERTScore comparison between summarization queries that model chose to answer vs. to reject.*
>
>     | BERTScore | 0.5B | 1.5B | 3B | 7B |
>     | --- | --- | --- | --- | --- |
>     | Rejected | 0.63 | 0.66 | 0.64 | 0.65 |
>     | Answered | 0.71 | 0.75 | 0.71 | 0.73 |
> - **Scenario Adaptability:** Crucially, the models maintained robust **Scenario Adaptability** (**Figure 11**). By adjusting the prompt, the models rationally adjusted their rejection rates to meet different cost constraints, mirroring their behavior on verifiable math tasks.
> In summary, these results demonstrate that DiSRouter's core mechanism—learning self-assessment via data-driven alignment—is not limited to exact-match tasks but is **transferable to open-ended generation** scenarios.

---

> > ### Author Response · Authors · 2025-11-26
> >
> > ## **W4: The evidence supporting the superiority of "intrinsic assessment" is marginal. In Table 5, the purpose-trained 7B DiSRouter (0.80 F1) barely outperforms a general-purpose 8B external classifier (0.77 F1). This negligible gap fails to provide a strong justification for the added complexity of the distributed training approach over a simpler, well-trained external router.**
> >
> > We respectfully disagree with the assessment that the performance gap is marginal. While the raw difference in F1 score on a single agent appears small, our new experiments demonstrate that this "marginal" advantage translates into a **substantial gain in system-level utility** due to the cumulative nature of the routing cascade.
> >
> > ### **1. Theoretical Clarification: Error Propagation in Cascades**
> >
> > Comparing single-node F1 scores in isolation underestimates the impact of precision in a distributed system. In a cascading architecture, errors are **cumulative**: a slight degradation in the rejection precision of early-stage models can lead to significant downstream inefficiency or accuracy loss.
> >
> > - **False Rejections:** A "marginal" increase in false rejections at early stages (small models) forces a disproportionately large number of simple queries to expensive downstream models, causing increased total cost.
> > - **False Acceptances:** A slight drop in rejection precision leads to immediate performance loss (incorrect answers).
> >
> > Therefore, what appears to be a 0.04 F1 gap in a single node can result in **a significant degradation of the global Utility metric**.
> >
> > ### **2. New Comparative Experiments**
> >
> > To rigorously validate this and isolate the benefit of "Intrinsic Assessment" from "Model Size" or "Architecture," we implemented two new strong baselines for the *Balance* scenario:
> >
> > - **`RouteLLM_large` (Centralized 8B):** A centralized router using the standard RouteLLM architecture (N-classification) but upgraded to use a powerful **Llama3-8B-Instruct** backbone. This controls for model size against our 7B agents.
> > - **`DiSRouter_external` (Cascade 8B):** A cascading system structurally identical to DiSRouter, but instead of using self-awareness, each agent is paired with a dedicated **external binary classifier** (also Llama3-8B-Instruct) to make the accept/reject decision. This isolates the "Intrinsic vs. External" variable.
> >
> > The results are summarized in **Table R6** below.
> >
> > *Table R6: Comparative ablation study isolating the impact of Routing Architecture (Centralized vs. Cascade) and Assessment Mechanism (External vs. Intrinsic) on System Utility.*
> >
> > | Index | Method | Architecture | Backbone Model for Router | Accuracy | Cost | Utility |
> > | --- | --- | --- | --- | --- | --- | --- |
> > | 1 | RouteLLM | centralized | RoBERTa-base | 0.44 | 0.16 | 0.36 |
> > | 2 | RouteLLM_large | centralized | Llama3-8B-Instruct | 0.57 | 0.37 | 0.39 |
> > | 3 | DiSRouter_external | cascading | Llama3-8B-Instruct | 0.57 | 0.29 | 0.42 |
> > | 4 | DiSRouter | cascading | LLM itself | 0.77 | 0.32 | 0.61 |
> >
> > ### **3. Analysis of Results**
> >
> > The comparison yields three critical insights:
> >
> > 1. **Impact of Router Size:** Comparing `RouteLLM_large` with the original `RouteLLM` (**Result 1 vs. 2**), we observe that while increasing the router size improves centralized performance, the gain is relatively limited.
> > 2. **Cascade vs. Centralized (External):** `DiSRouter_external` outperforms `RouteLLM_large` (**Result 2 vs. 3**). This confirms that a cascading architecture—where classifiers focus on the simpler task of modeling a single agent's binary boundary—is easier to optimize than a centralized router attempting to rank all agents simultaneously.
> > 3. **Intrinsic vs. External (The Critical Gap):** Most importantly, `DiSRouter (Intrinsic)` significantly outperforms `DiSRouter_external` (**Result 3 vs. 4**, Utility **0.61 vs. 0.42**).
> >
> > **Conclusion:** This substantial gap in Utility (**+45% relative improvement**) decisively validates our approach. It proves that an external judge—even a powerful 8B model—cannot perfectly align with the target model's internal knowledge boundary. **Intrinsic self-awareness** eliminates this alignment mismatch, providing a superior routing signal that justifies the training design.

---

### Official Review · Reviewer_qdad · 2025-10-30

**Soundness:** 2
**Presentation:** 4
**Contribution:** 2
**Rating:** 4
**Confidence:** 2

**Summary:**

This paper proposes DiSRouter—a distributed self-routing framework that can propagate queries step-by-step through different LLM agents without a central router.
A two-stage Self-Awareness Training (SFT+RL) is designed to teach individual LLMs to "answer if they know the answer and reject if they don't," and the rejection threshold can be adaptively adjusted according to user preferences (performance priority/balance/cost priority).

**Strengths:**

1.Experiments show that DiSRouter's utility score (performance-α·cost) is significantly superior to both baselines and existing router systems across various performance-cost scenarios, with both in-domain and out-of-domain data. It is more flexible and has "plug-and-play" modular capabilities.

2.The hypothesis that training does not inject new knowledge but only improves self-boundary awareness can be supported by experimental results: ∆Performance <1%, which verifies that the performance improvement mainly comes from better routing rather than the model itself becoming stronger.

**Weaknesses:**

Although the experimental results have shown that the authors' method is far superior to the baseline and other routing methods, I believe that some additional perspectives are still beneficial for future work:

1.The cost introduced by the two-stage training approach may require further evaluation or comparison with other methods, as each agent needs to perform independent multiple-time inference to prepare its SFT data and conduct RL training.

2.For more open agent systems, data preparation is more challenging, as some questions do not have a single correct answer (e.g., summarizing, creation/generation questions) while still have varying difficulty, making the self-awareness training phase tricky. If only verifiable questions data are synthesized and trained, unprepared tasks may suffer from catastrophic forgetting, and the model's self-awareness on such tasks cannot be guaranteed. The author may provide the performance of the model and the systems on these more complex out-of-domain tasks.

**Questions:**

1.In Section 3.2.2 and Appendix C, what is the purpose of extracting and reformulating the original model's answer using a stronger model? The intervention of the model may introduce additional errors and information, resulting in unpredictable performance after training.

2.Is the true "routing overhead" underestimated? There may be multiple levels of consecutive rejections on the cascading path (the experiment used a 5-model approach, and the worst case was only 4 rejections to reach 14 bytes). Could you please provide the end-to-end latency of the first token for comparison with other methods?

---

> ### Author Response · Authors · 2025-11-26
>
> ## **W1: The cost introduced by the two-stage training approach may require further evaluation or comparison with other methods, as each agent needs to perform independent multiple-time inference to prepare its SFT data and conduct RL training.**
>
> We thank the reviewer for this detailed examination of the computational costs associated with our two-stage training approach. While our method involves independent SFT and RL phases for each agent, a deeper analysis reveals that the **comparative overhead is marginal** in the short term and **highly cost-effective** in the long term.
>
> ### **1. Comparable Data Preparation and Training Efficiency**
>
> The perceived overhead of our data generation phase is mitigated by modern inference acceleration frameworks.
>
> - **Data Sampling:** Centralized router baselines also require a data sampling phase to acquire ground-truth labels for training. By leveraging acceleration frameworks like vLLM, the time difference becomes negligible. For instance, sampling the complete dataset for our largest agent (7B) requires only **~1.5 hours**.
> - **Training Time vs. Scenario Adaptability:** While our 7B model requires approximately **2 hours** for training (higher than a lightweight centralized router), this cost is offset by our **Scenario Adaptability**. A single DiSRouter agent, once trained, can adapt to various deployment constraints (e.g., *Performance First*, *Balance*, *Cost First*) simply via prompt adjustments. In contrast, centralized baselines typically require **retraining the router from scratch three separate times** to optimize for these different reward functions, thereby introducing multiplicative training costs that rival or exceed our own.
>
> ### **2. Amortized Overhead via "Plug-and-Play" Modularity**
>
> As detailed in **Table 4 in the paper**, the training cost in DiSRouter is a **one-time investment** per agent due to its distributed nature.
>
> - **Decoupled Maintenance:** Once an agent is trained, it becomes a modular component that can be reused indefinitely. If the agent pool changes (e.g., adding a new model), DiSRouter only requires training the *new* agent.
> - **Baseline Inefficiency:** Conversely, centralized routers are coupled to the specific pool; any modification to the model pool necessitates a complete **retraining of the entire routing model**. Thus, in dynamic industrial environments, DiSRouter offers significantly better amortized cost-efficiency.
>
> ### **3. Training is Optional for High-Capacity Models**
>
> Finally, we emphasize that Self-Awareness Training is **not mandatory** for all models as discussed in **Appendix A.9**.
>
> - **Intrinsic Self-Awareness:** As shown in **Appendix A.9.2**, SOTA models (e.g., `GPT` series) possess strong intrinsic self-awareness. Our experiments confirm they can exhibit rational rejection behavior and distinguish solvable queries via prompting alone (see **Figure 9** and **Table R1** for "Training-Free" results).
>
>     *Table R1: GPT models have an **inherent capacity for self-assessment**, showing varied accuracy for answered queries vs. rejected queries.*
>
>     | Split | GSM8K | MATH-500 | AIME24 | AIME25 | AMC23 | JEEBench | OlympiadBench | OlympicArena |
>     | --- | --- | --- | --- | --- | --- | --- | --- | --- |
>     | Answered | 0.91 | 0.79 | 0.5 | 0.14 | 0.75 | 0.52 | 0.67 | 0.57 |
>     | Rejected | 0.79 | 0.62 | 0.18 | 0.22 | 0.56 | 0.38 | 0.32 | 0.15 |
> - **Training-Free DiSRouter:** Leveraging this finding, we constructed a **Training-Free DiSRouter** using three GPT models (GPT4.1-nano, GPT4.1-mini, and o4-mini). This system achieved utility comparable to router baselines on in-domain tasks and **significantly outperformed them on out-of-domain (OOD) tasks** where baselines often failed to generalize (**Table R2**).
>
>     *Table R2：Utility of routing systems on in-domain and out-of-domain tasks.*
>
>     |  |  | ID |  |  | OOD |  |
>     | --- | --- | --- | --- | --- | --- | --- |
>     | Method | Accuracy | Cost | Utility | Accuracy | Cost | Utility |
>     | Oracle | 0.97 | 0.18 | 0.88 | 0.82 | 0.48 | 0.58 |
>     | Random | 0.89 | 0.49 | 0.65 | 0.62 | 0.53 | 0.36 |
>     | GraphRouter | 0.92 | 0.4 | 0.72 | 0.57 | 0.39 | 0.38 |
>     | DiSRouter | 0.84 | 0.17 | 0.76 | 0.61 | 0.41 | 0.41 |
> - **Future Outlook:** This suggests that as open-source models improve, the need for explicit training will diminish, allowing them to be integrated into DiSRouter via prompt engineering, further eliminating training overhead.
>
> In summary, while the per-agent training setup appears computationally heavier upfront, the **adaptability across scenarios**, **modularity in maintenance**, and **optionality for advanced models** make DiSRouter a highly efficient solution for long-term deployment.

---

> > ### Author Response · Authors · 2025-11-26
> >
> > ## **W2: Adaptation to open-ended tasks that do not have a single correct answer (e.g., summarizing, creation/generation questions).**
> >
> > We thank the reviewer for highlighting the importance of **open-ended tasks**, which are indeed critical for evaluating broader generalization. We have extended our framework to address such tasks and conducted new generalization experiments on **text summarization**. The results confirm that our **Self-Awareness Training Pipeline generalizes effectively**, enabling models to distinguish high-quality generation from low-quality generation even when the metric is continuous rather than binary.
> >
> > We have included the full experimental setup and detailed results in **Appendix A.11** of the revised paper.
> >
> > ### **1. Methodological Adaptation: From Binary to Continuous Scoring**
> >
> > As detailed in **Appendix A.11.1**, we adapted our pipeline to handle the continuous nature of open-ended tasks (where quality is not a simple $\{0, 1\}$):
> >
> > **Percentile-based SFT Threshold:** Unlike verifiable tasks with binary outcomes, summarization quality (e.g., BERTScore) is continuous. We replaced the fixed probability threshold with a **percentile-based threshold**. Specifically, the rejection threshold $\delta$ is dynamically set to the $(1-\alpha)\text{-th}$ percentile of the model's score distribution on the training set.
> >
> > **Continuous RL Reward:** Similarly, the reward for answering is defined as the continuous score output by the evaluation metric. The reward for rejecting is anchored to the $(1-\alpha)\text{-th}$ percentile baseline. This effectively normalizes the reward signal, allowing the model to learn relative quality distinctions.
> >
> > ### **2. Experimental Validation on Summarization**
> >
> > We evaluated this approach on two standard datasets, **SAMSum** and **XSum**, using **BERTScore** as the quality metric (**Appendix A.11.2**).
> >
> > - **Strong Discriminative Capability:** Post-alignment, the models demonstrated significant self-awareness. The average BERTScore of queries the model chose to Answer was significantly higher than that of the queries it Rejected (see **Table R3 and Figure 12**), proving it can successfully identify its own high-quality outputs.
> >
> >     *Table R3: BERTScore comparison between summarization queries that model chose to answer vs. to reject.*
> >
> >     | BERTScore | 0.5B | 1.5B | 3B | 7B |
> >     | --- | --- | --- | --- | --- |
> >     | Rejected | 0.63 | 0.66 | 0.64 | 0.65 |
> >     | Answered | 0.71 | 0.75 | 0.71 | 0.73 |
> > - **Scenario Adaptability:** Crucially, the models maintained robust **Scenario Adaptability** (**Figure 11 in the paper**). By adjusting the prompt, the models rationally adjusted their rejection rates to meet different cost constraints, mirroring their behavior on verifiable math tasks.
> >
> > In summary, these results demonstrate that DiSRouter's core mechanism—learning self-assessment via data-driven alignment—is not limited to exact-match tasks but is **transferable to open-ended generation** scenarios.
> >
> > ## **Q1: In Section 3.2.2 and Appendix C, what is the purpose of extracting and reformulating the original model's answer using a stronger model?**
> >
> > We appreciate the opportunity to clarify this design choice regarding **Data Construction** (**Section 3.2.2**). The primary motivation for utilizing a stronger model (14B) to extract and reformulate the Chain-of-Thought (CoT) is **Data Cleaning and Standardization**, rather than knowledge augmentation.
> >
> > ### **1. Addressing Weak Instruction Following in Small LLMs**
> >
> > While rule-based extraction is a theoretical alternative, our preliminary experiments revealed that smaller LLMs (e.g., 0.5B) often exhibit **weak instruction-following capabilities** regarding strict output formatting. They frequently fail to generate consistent markers (e.g., specific prefixes or separators) required for accurate parsing. Using a stronger LLM as an extractor ensures a **standardized data format** for the SFT phase, allowing the target model to focus its capacity on learning the core self-assessment logic rather than struggling with noisy or inconsistent syntax.
> >
> > ### **2. Minimal Risk of Hallucination**
> >
> > Regarding the concern about introducing external errors:
> >
> > - **Task Simplicity:** The task assigned to the 14B model is simply **extraction and formatting**, not reasoning generation.
> > - **Manual Verification:** We conducted a manual inspection of sampled cases, which confirmed that the extraction process is precise. The 14B model successfully isolates the original CoT reasoning **without hallucinating** new steps or introducing additional information.
> >
> > Therefore, this process acts as a robust noise-filtering step, ensuring high-quality training data while minimizing the risk of distribution shift or artificial performance inflation.

---

> > > ### Author Response · Authors · 2025-11-26
> > >
> > > ## **Q2: Is the true "routing overhead" underestimated? There may be multiple levels of consecutive rejections on the cascading path (the experiment used a 5-model approach, and the worst case was only 4 rejections to reach 14 bytes). Could you please provide the end-to-end latency of the first token for comparison with other methods?**
> > >
> > > We appreciate the reviewer's scrutiny regarding the **practical latency** of our cascading architecture. We share the concern that multiple consecutive rejections could theoretically accumulate delay. However, our new comprehensive analysis in **Appendix A.1** empirically demonstrates that this overhead is **statistically negligible**.
> > >
> > > ### **1. Negligible Cumulative Overhead (<5%)**
> > >
> > > Our data confirms that the "routing overhead" (time spent on rejections) is marginal compared to the cost of generation.
> > >
> > > - **Worst-Case Analysis:** Even in the worst-case path—where a query is rejected by all four smaller agents before reaching the final 14B model—the cumulative time consumed by these rejections constitutes **less than 5%** of the time required for the final LLM to generate the answer (*see* **Table 7 in the paper**).
> > > - **Reasoning:** This efficiency stems from the **brevity of the rejection signal**. A rejection typically involves generating only a few tokens (e.g., "I don't know!"), whereas generating a valid solution involves a lengthy CoT sequence. Thus, the cost of “saying no" is a fraction of the cost of solving.
> > >
> > > ### **2. Time-to-First-Token Analysis**
> > >
> > > To provide the requested comparison, we benchmarked the **System Latency (Time to First Token)** of DiSRouter against router baselines (detailed in **Table R4**).
> > >
> > > *R4: Comparison of Time to First Token (s) between routing systems.*
> > >
> > > |  | RouterLLM | FrugalGPT | Automix | FORC | GraphRouter | DiSRouter |
> > > | --- | --- | --- | --- | --- | --- | --- |
> > > | Average | 0.048 | 2.620 | 3.475 | 0.070 | 0.024 | 0.081 |
> > > | Worst Case | 0.063 | 9.565 | 11.34 | 0.076 | 0.028 | 0.241 |
> > > - **vs. Predictive Routers:** DiSRouter incurs slightly higher latency than predictive routers (e.g., *RouteLLM*, *GraphRouter*) that utilize lightweight external classifiers, as our method requires LLM inference.
> > > - **vs. Generative Baselines:** Crucially, DiSRouter is **significantly faster** than other generative or verification-based routers (e.g., *FrugalGPT*, *AutoMix*). These baselines often require generating **full-length responses** from LLMs before performing verification or cascading, whereas DiSRouter aborts generation immediately upon detecting incompetence.
> > >
> > > **Conclusion:** Based on this analysis, we conclude that the "routing overhead" introduced by the rejection mechanism is **computationally minor** and does not impede the system's practical utility or user experience.

---

### Official Review · Reviewer_xhvr · 2025-10-31

**Soundness:** 3
**Presentation:** 3
**Contribution:** 3
**Rating:** 6
**Confidence:** 3

**Summary:**

This paper introduces **DiSRouter (Distributed Self-Router)**, a framework that replaces centralized query routing in LLM systems with a distributed paradigm where each agent autonomously decides to answer or route queries based on self-awareness. Key contributions include:

- A **distributed routing architecture** that enhances modularity and scalability compared to centralized routers.
- A **Self-Awareness Training pipeline** with supervised fine-tuning (SFT) and reinforcement learning (RL) using a localized reward function, enabling independent agent training.
- **Scenario adaptability** via a preference factor $\alpha$ to balance performance and cost dynamically. Experiments on datasets like GSM8K and MMLU show DiSRouter outperforms baselines (e.g., RouteLLM, FrugalGPT) in utility. The core idea is that intrinsic self-assessment is more effective than external evaluation.

**Strengths:**

- **Originality**: The shift from centralized to distributed routing is novel, creatively integrating self-assessment concepts (e.g., LLM uncertainty) into a routing framework. The scenario-adaptive reward function is an innovative touch.
- **Quality**: Experimental design is rigorous within its scope, with utility metrics and modularity tests. The pipeline is methodically described.
- **Clarity**: The problem formulation in Section 2 is precise. Writing is concise.
- **Significance**: The "plug-and-play" modularity addresses a key pain point in LLM systems, though real-world impact depends on scalability beyond the tested settings.

**Weaknesses:**

- **Limited Generalizability**: Experiments use only the Qwen2.5-Instruct series, neglecting other LLM families (e.g. Llama). This casts doubt on DiSRouter's applicability to diverse models. The authors should include cross-architecture validation.
- **Training Efficiency Concerns**: The Self-Awareness Training (SFT + RL) is computationally heavy, but the paper dismisses routing cost as "negligible" without quantifying training overhead. A cost-benefit analysis is missing.
- **Limited Robustness Testing**: There is no evaluation under adversarial conditions (e.g., ambiguous queries) to test the robustness of self-assessment. This is critical for real-world reliability.

**Questions:**

1. I am still confused that how the the local routing policy $\pi_i$ of agent $m_i$ is trained, it seems that $m_i$ only output "I don't know!" if choosing reject, how does $m_i$ select the next agent?
2. Can DiSRouter's self-awareness training be applied to LLMs with different architectures (e.g., encoder-decoder models)? Please provide results with at least one non-Qwen model family to validate broad applicability.
3. How does DiSRouter perform with large-scale agent pools (e.g., >10 agents)?
4. What are the practical limitations (e.g., latency, fault tolerance) in deploying DiSRouter? Case studies or simulations with real-world constraints would be valuable.

---

> ### Author Response · Authors · 2025-11-26
>
> ## **W1: Limited Generalizability: Experiments use only the Qwen2.5-Instruct series, neglecting other LLM families (e.g. Llama). This casts doubt on DiSRouter's applicability to diverse models. The authors should include cross-architecture validation.**
>
> We are grateful for this insightful comment concerning the **cross-architecture applicability** of DiSRouter. Our new experiments confirm the strong generalization capability of both the Self-Awareness Training Pipeline and the overall DiSRouter architecture.
>
> ### **Experimental Validation of Cross-Architecture Generalization**
>
> To rigorously validate this, we conducted a new set of experiments that integrated models from distinct families into our system.
>
> 1. **Agent Integration:** We applied our Self-Awareness Training to two new models, **Gemma3-1B-Instruct** (1B) and **Phi4-mini-Instruct** (4B), to enhance their rejection capabilities.
> 2. **System Modification:** Leveraging the **"plug-and-play" modularity** of DiSRouter, we modified our original five-agent routing system (0.5B-1.5B-3B-7B-14B) by removing the three smallest models and substituting them with these two new models. This resulted in a **heterogeneous four-agent routing system (1B-4B-7B-14B)**, which was then benchmarked against other router baselines.
>
> The results decisively confirm the generalizability across different model families:
>
> 1. **Effective Adaptation:** Post-training, both the Gemma3 and Phi4 models demonstrated **strong self-assessment capabilities** and **Scenario Adaptability**, successfully distinguishing in-competence queries and adhering to cost constraints. (**Table R1**)
>
>     *Table R1: A summary of model self-awareness ability in the Balance scenario. $Ratio_{ans}$ refer to the answer rate, $Acc_{ans}$ and $Acc_{rej}$ refer to the accuracy on the subset of queries the model chose to answer and reject, respectively.*
>
>     | Backbone Model | Ratio$_{ans}$ | Acc$_{ans}$ | Acc$_{rej}$  |
>     | --- | --- | --- | --- |
>     | Gemma-3-1B-instruct | 8.82 | 76.66 | 38.31 |
>     | Phi-4-mini-instruct | 72.07 | 85.64 | 56.62 |
>
> 2. **Superior Utility:** The resulting heterogeneous DiSRouter system continued to achieve significantly superior routing Utility, delivering higher performance at a lower computational cost compared to centralized router baselines. (**Table R2**)
>
>     *Table R2: Utility of routing systems.*
>
>     | Method | Accuracy | Cost | Utility |
>     | --- | --- | --- | --- |
>     | Oracle | 0.94 | 0.41 | 0.74 |
>     | Random | 0.71 | 0.56 | 0.43 |
>     | GraphRouter | 0.81 | 0.7 | 0.46 |
>     | DiSRouter | 0.80 | 0.53 | 0.54 |
>
>     Full details could be found in Tables 19 and 20 of the revised paper.
>
> ### **Validation of Distributed Design**
>
> Crucially, this experiment further underscores the key advantage of DiSRouter’s distributed design: modifying the agent pool only required the **one-time training** of the new agents (Gemma3 and Phi4). The existing agents (Qwen2.5-Instruct 7B/14B) were **reused directly without any retraining**, validating the system's long-term maintainability and scalability.
>
> We have included the complete experimental setup and detailed results in **Appendix A.10 of the revised manuscript** and encourage the reviewer to examine them.

---

> > ### Author Response · Authors · 2025-11-26
> >
> > ## **W2: Training Efficiency Concerns: The Self-Awareness Training (SFT + RL) is computationally heavy, but the paper dismisses routing cost as "negligible" without quantifying training overhead. A cost-benefit analysis is missing.**
> >
> > We appreciate the reviewer's attention to the crucial matter of training efficiency and cost-benefit analysis. While we acknowledge that the initial Self-Awareness Training for a single agent is computationally heavier than training a lightweight centralized router, our detailed analysis (now in **Appendix A.9**) demonstrates that the overall and long-term costs are competitive or superior due to the **amortized and optional nature** of our training pipeline.
> >
> > ### **1. Self-Awareness Training is not always mandatory but optional**
> >
> > The training pipeline is not a universal requirement but an enhancement for models that lack intrinsic self-awareness (**Appendix A.9.1**).
> >
> > - **Intrinsic Self-Awareness:** For foundation models with **sufficient intrinsic self-awareness** (such as the GPT series), the training can be entirely bypassed, with the agent integrated via simple **prompt engineering**. We experimentally validated the strong intrinsic self-awareness of the GPT series (**Appendix A.9.2**), showing that they exhibit:
> >     - **Rational Rejection Behavior:** A strong positive correlation between their answer rate and their actual capability on complex tasks (**Figure 9**).
> >     - **Distinguished Solvable Queries:** Significantly higher accuracy on answered samples compared to rejected samples (**Table R3**).
> >
> >     *Table R3: GPT models have an inherent capacity for self-assessment, showing varied accuracy for answered queries vs. rejected queries.*
> >
> >     | Split | GSM8K | MATH-500 | AIME24 | AIME25 | AMC23 | JEEBench | OlympiadBench | OlympicArena |
> >     | --- | --- | --- | --- | --- | --- | --- | --- | --- |
> >     | Answered | 0.91 | 0.79 | 0.5 | 0.14 | 0.75 | 0.52 | 0.67 | 0.57 |
> >     | Rejected | 0.79 | 0.62 | 0.18 | 0.22 | 0.56 | 0.38 | 0.32 | 0.15 |
> >
> > - **Training-Free DiSRouter:** Leveraging this finding, we constructed a Training-Free DiSRouter using three GPT models (GPT4.1-nano, GPT4.1-mini, and o4-mini). This system achieved utility comparable to router baselines on in-domain tasks and **significantly outperformed them on out-of-domain (OOD) tasks** where baselines often failed to generalize (**Table R4**).
> >
> >     *Table R4：Utility of routing systems on in-domain and out-of-domain tasks.*
> >
> >     |  |  | ID |  |  | OOD |  |
> >     | --- | --- | --- | --- | --- | --- | --- |
> >     | Method | Accuracy | Cost | Utility | Accuracy | Cost | Utility |
> >     | Oracle | 0.97 | 0.18 | 0.88 | 0.82 | 0.48 | 0.58 |
> >     | Random | 0.89 | 0.49 | 0.65 | 0.62 | 0.53 | 0.36 |
> >     | GraphRouter | 0.92 | 0.40 | 0.72 | 0.57 | 0.39 | 0.38 |
> >     | DiSRouter | 0.84 | 0.17 | 0.76 | 0.61 | 0.41 | 0.41 |
> >
> > This verifies that **Self-Awareness Training is an enhancement, not a strict requirement**, provided the base LLM possesses a high degree of intrinsic self-assessment capability. We believe this self-awareness will become a standard criterion in future SOTA models.
> >
> > ### **2. Amortized Training Overhead in the Long Run**
> >
> > The training overhead is fully **amortized** over the system's lifetime due to the decoupled nature of DiSRouter.
> >
> > - **Decoupled Training:** The training for each agent is a **one-time investment**, which, once completed, allows the agent to be a modular, "plug-and-play" component that can be **flexibly reused across different routing systems and tasks** ("train once, run anywhere"). This is reinforced by the experiment in our response to `W1` and **Table 4**.
> > - **Centralized Router Cost:** In contrast, centralized router baselines, while having faster initial training for a fixed pool, necessitate a complete and expensive **retraining of the entire router** for *any* modification to the agent pool or the underlying tasks.
> >
> > This architectural advantage grants DiSRouter a significant long-term **cost-effectiveness and operational flexibility**, making the cumulative cost comparable or superior in dynamic, real-world deployment scenarios.
> >
> > We believe this detailed analysis fully addresses the reviewer's concern, and we invite them to review the full details in **Appendix A.9**.

---

> > > ### Author Response · Authors · 2025-11-26
> > >
> > > ## **W3: Limited Robustness Testing: There is no evaluation under adversarial conditions (e.g., ambiguous queries) to test the robustness of self-assessment. This is critical for real-world reliability.**
> > >
> > > We sincerely thank the reviewer for highlighting the necessity of evaluating the robustness of self-assessment under adversarial or **ambiguous conditions**, which is critical for real-world reliability. Our new analysis demonstrates that the training pipeline successfully sharpens the model's knowledge boundary, enabling **robust self-assessment** and promoting **conservative behavior** even on highly uncertain inputs.
> > >
> > > ### **Analytical Validation on Ambiguous Queries**
> > > To address this, we conducted an analytical validation specifically focusing on model behavior when faced with **"ambiguous queries."**
> > >
> > > 1. **Defining Ambiguity:** We defined ambiguous queries as those lying in the "**fuzzy region**" of the model's knowledge boundary—queries where the base LLM achieved approximately 50% correctness across 10 stochastic sampling rollouts.
> > > 2. **Test Set Construction:** Following the methodology in **Section 4.3**, we measured self-assessment as a binary classification problem. We constructed two balanced test sets for the 7B agent:
> > >     1. **Ambiguous Set:** Queries with sampling accuracy $\approx 0.5$.
> > >     2. **Deterministic Set:** Queries where the model was highly certain (sampling accuracy $>0.8$ or $<0.2$).
> > >
> > > ### **Results and Analysis**
> > >
> > > The classification performance on these sets confirms the robustness of our approach (**Table R5**):
> > >
> > > *Table R5: Performance of the Self-awareness Trained LLM (7B) across Different Confidence Splits.*
> > >
> > > | Split | Accuracy | F1 | Precision | Recall |
> > > | --- | --- | --- | --- | --- |
> > > | Full | 0.80 | 0.81 | 0.76 | 0.86 |
> > > | Ambiguous Set | 0.66 | 0.62 | 0.73 | 0.54 |
> > > | Deterministic Set | 0.88 | 0.89 | 0.88 | 0.91 |
> > > 1. **Discriminative Ability on Ambiguity:** On highly ambiguous queries, the model still maintained discriminative power, achieving **Accuracy and F1 scores exceeding the random baseline of 0.5**. This suggests that our Self-Awareness Training effectively **sharpens the model's knowledge boundary**, enabling it to make meaningful distinctions even on uncertain samples.
> > > 2. **Conservative Behavior:** Notably, on ambiguous queries, the model exhibited high Precision but lower Recall (specifically, for the "answerable" class). This indicates a **conservative tendency**: when faced with significant ambiguity, the model prefers to reject (resulting in lower recall on these uncertain, answerable samples) rather than risk providing an incorrect answer. We consider this a highly desirable trait for a reliable system, as it actively **minimizes potential hallucination** and ensures system dependability.
> > > 3. **High Performance on Certainty:** On deterministic queries, the self-assessment achieved **exceptional Accuracy and F1 scores**, further validating the effectiveness of leveraging intrinsic knowledge for routing decisions.
> > >
> > > In conclusion, our method demonstrates robust and reliable self-assessment performance, particularly exhibiting a **risk-averse behavior** on ambiguous inputs, which enhances the overall trustworthiness of the DiSRouter system in real-world scenarios.
> > >
> > > ## **Q1 I am still confused that how the the local routing policy pi_i of agent m_i is trained, it seems that m_i only output "I don't know!" if choosing reject, how does m_i select the next agent?**
> > >
> > > We apologize for the confusion regarding the local routing policy $\pi_i$ and thank the reviewer for asking for clarification.
> > >
> > > ### **Binary Action Space and Deterministic Cascade**
> > >
> > > In our current implementation, DiSRouter is structured as a **sequential cascade** of agents, ordered by increasing size and cost (as illustrated in **Figure 2** of the main paper).
> > >
> > > 1. **Binary Action Space:** The local routing policy $\pi_i$ of agent $m_i$ possesses a simple binary action space: it can either **execute the query** (by generating a final answer) or **reject the query** (by outputting the predefined refusal phrase, "I don't know!").
> > > 2. **Deterministic Routing:** If agent $m_i$ chooses to reject a query, the system **deterministically routes** the query to the **immediate next agent** ($m_{i+1}$) in the predefined cascade sequence.
> > >
> > > Therefore, agent $m_i$ **does not need to explicitly select a specific target** from the remaining pool; the "next agent" is structurally and deterministically pre-defined by the system's cascade design. The policy $\pi_i$ is solely responsible for the binary decision of "Answer vs. Reject" based on the agent's self-assessed competence.
> > > We acknowledge that more complex topologies (such as tree or graph structures) would require explicit next-agent selection, and we are currently exploring these as important future work.

---

> > > > ### Author Response · Authors · 2025-11-26
> > > >
> > > > ## **Q2: Can DiSRouter's self-awareness training be applied to LLMs with different architectures (e.g., encoder-decoder models)? Please provide results with at least one non-Qwen model family to validate broad applicability.**
> > > >
> > > > As detailed in our Response to `W1`, our method has proven effective across distinct model families (Gemma3, Phi4). We prioritized decoder-only architectures in our experiments simply because they currently dominate the high-performance LLM landscape, leaving few competitive encoder-decoder options for complex reasoning.
> > > >
> > > > Crucially, our Self-Awareness Training is a **data-driven alignment strategy** (SFT + RL) rather than an architecture-specific modification. Consequently, the methodology is architecture-agnostic and should be applicable to any model architecture—including encoder-decoder frameworks—provided the model possesses sufficient capacity to learn self-reflection from data.
> > > >
> > > > ## **Q3: How does DiSRouter perform with large-scale agent pools (e.g., >10 agents)?**
> > > >
> > > > In our current experimental setup, we utilized a pool of **general-purpose LLMs** (`Qwen2.5-Instruct` series). Since these models are homogeneous and possess broad capabilities across various domains, a concise cascade of 5 agents proved sufficient to effectively cover the difficulty spectrum of the test tasks without requiring a large-scale pool (>10 agents).
> > > >
> > > > However, we are actively researching **heterogeneous routing networks** for future work. In such scenarios, the pool would contain diverse, domain-specific expert models (e.g., **specialized math solvers or coding experts**). This setting would naturally benefit from a larger scale DiSRouter network, likely organized in more complex topologies such as **tree structures** rather than a simple linear cascade. As this paper serves as the foundational work validating the efficacy of the distributed self-routing paradigm, we have focused on the cascade structure to establish the core concept. We intend to explore large-scale, heterogeneous agent pools in subsequent research.
> > > >
> > > > ## **Q4: What are the practical limitations (e.g., latency, fault tolerance) in deploying DiSRouter? Case studies or simulations with real-world constraints would be valuable.**
> > > >
> > > > We appreciate this practical perspective, which was echoed by other reviewers. To address concerns regarding latency and overhead, we have added a detailed statistical analysis of actual inference and routing costs in **Appendix A.1** of the revised paper.
> > > >
> > > > ### **Latency Analysis**
> > > >
> > > > 1. **Negligible Routing Overhead:** Our empirical data (**Table 6**) shows that although a query may be rejected by multiple agents before being solved, the cumulative **routing cost** (time spent generating rejections) remains negligible. Even in the worst-case scenario (routing to the final agent), this overhead constitutes less than **5%** of the actual inference cost of generating the final answer (**Table 7**).
> > > > 2. **Competitive Latency:** We also benchmarked **System Latency** (Time to First Token, **Table R6 and Table 8 in paper**). Results indicate that while DiSRouter incurs slightly higher latency than predictive routers (e.g., RouteLLM) which use lightweight classifiers, it is significantly faster than generative baselines that rely on full model outputs or verification steps (e.g., FrugalGPT, AutoMix).
> > > >
> > > >     *Table R6: Time to First Token (s) Results Comparison for Different Routing Systems.*
> > > >
> > > >     |  | RouterLLM | FrugalGPT | Automix | FORC | GraphRouter | DiSRouter |
> > > >     | --- | --- | --- | --- | --- | --- | --- |
> > > >     | Average | 0.048 | 2.620 | 3.475 | 0.070 | 0.024 | 0.081 |
> > > >     | Worst Case | 0.063 | 9.565 | 11.34 | 0.076 | 0.028 | 0.241 |
> > > >
> > > > 3. **Deployment Constraints:** We acknowledge that our current evaluation assumes efficient inter-agent communication. In **physically distributed deployments**, **network latency** becomes a non-negligible factor. Real-world implementation would require incorporating network transmission overhead into the global cost function to optimize for total system latency.

---

### Author Response · Authors · 2025-12-02
**Summary of Reviews & New Experiments: Addressing All Reviewer Concerns with Definitive Evidence**

**Dear AC,**

Thank you for accepting our paper as a re-assigned submission. We truly appreciate the additional workload this may have caused. We are also grateful for the reviewers’ thoughtful and constructive feedback.

**We appreciate that the reviewers acknowledged the contributions of our work:**

- **Reviewer JRcd** highlighted that the problem formulation is clear and that our **"idea will be of interest to the community".**
- **Reviewer xhvr** praised the "originality" of shifting from centralized to distributed routing and noted the "rigorous" experimental design.
- **Reviewer qdad** recognized that DiSRouter achieves a utility score "significantly superior" to baselines and appreciated its "plug-and-play" modularity.
- **Reviewer dfTT** acknowledged the "concrete technical contribution" of our Self-Awareness Training pipeline.

---
### **Summary of Key Concerns and Our Responses**
We consolidated the feedback into four core questions and present our responses below.

**Core Concern A: Cross-Architecture Generalization (Homogeneity)**

*(Raised by Reviewers xhvr [W1], dfTT [W2])*

**Reviewer Comment:** Concerns were raised that our experiments relied solely on the Qwen series, questioning applicability to heterogeneous agent pools.

**Our Response:** We conducted **new cross-architecture experiments** (as presented in Appendix A.10 of the revised paper) by integrating `Gemma3-1B-Instruct` and `Phi4-mini-Instruct` into the routing chain. The results confirm that our Self-Awareness Training is **architecture-agnostic**. The heterogeneous system (Gemma/Phi/Qwen) maintained high scenario adaptability and superior utility, demonstrating that DiSRouter generalizes effectively to diverse model families.

---
**Core Concern B: Routing Overhead and Utility Metric Formulation**

*(Raised by Reviewers dfTT [W1], qdad [Q2], JRcd [W2])*

**Reviewer Comment:** Reviewer dfTT argued that our Utility metric was flawed for ignoring cumulative latency (routing overhead) in the cascade. Reviewers qdad and JRcd are also concerned about the overhead of multiple rejection responses.

**Our Response:** We clarified that the omission of the routing overhead term in the original formula was a **deliberate simplification for mathematical conciseness**, predicated on our analysis that this term is statistically negligible as presented in Appendix A.1. To validate this:

1. **Empirical Verification:** We measured that the cumulative "routing cost" (time spent rejecting) constitutes **<5%** of the inference cost even in worst-case paths.
2. **Adjusted Metric:** We re-calculated our results using a stricter Utility metric that explicitly subtracts routing overhead ($Cost_{routing}$). As shown in Table R2 (Response to dfTT), the utility drop is marginal (~1.5%) and DiSRouter retains its superiority over baselines.
3. **Time-to-First-Token:** We demonstrated that while DiSRouter incurs slightly higher latency than predictive routers (e.g.,s RouteLLM) which use lightweight classifier, it is **significantly faster** than generative baselines that rely on full model outputs or verification steps (e.g., FrugalGPT, AutoMix).

---
**Core Concern C: Training Efficiency and Overhead**

*(Raised by Reviewers xhvr [W2], qdad [W1])*

**Reviewer Comment:** Reviewers expressed concern that the two-stage training approach (SFT+RL) is computationally heavy compared to training a lightweight centralized router.

**Our Response:** We argue that the training overhead is justified and optional:

1. **Optionality (GPT Series Experiments):** We demonstrated that high-capability models (e.g., GPT-4o-mini) possess **intrinsic self-awareness** and do not require training. We constructed a **"Training-Free DiSRouter"** using GPT models, which achieved comparable utility on in-domain tasks and much higher utility on OOD tasks without any gradient updates.
2. **Amortized Cost:** Unlike centralized routers that require full retraining whenever the model pool and tasks change, DiSRouter is decoupled. Training is a one-time investment per agent; once trained, an agent is a modular "plug-and-play" component.
---
**Core Concern D: Generalization to Open-Ended Tasks**

*(Raised by Reviewers dfTT [W3], qdad [W2])*

**Reviewer Comment:** Questions regarding whether the binary "reject/answer" training applies to open-ended tasks like summarization where correctness is non-binary.

**Our Response:** We extended our framework to **Text Summarization** (SAMSum, XSum) in Appendix A.11. We adapted the reward function to use continuous metrics (BERTScore) and percentile-based thresholds. The results show that DiSRouter successfully distinguishes high-quality generations from low-quality ones, proving the method is **transferable to open-ended generation tasks.**

---

> ### Author Response · Authors · 2025-12-02
>
> ### **Conclusion**
>
> With these additional validations, our work presents a comprehensive contribution:
>
> 1. **A Novel Distributed Routing Framework:** We introduce the "Distributed Self-Router" framework, which leverages **intrinsic self-assessment** for routing decisions. This approach resolves the inaccuracy of external evaluations and the inflexibility inherent in traditional centralized routers, offering a more principled and effective **paradigm for query routing** tasks.
> 2. **An Effective Self-Awareness Training Pipeline:** We propose a robust two-stage training pipeline (SFT+RL) that effectively enhances the **self-assessment capability and reliability** of open-source LLMs.
> 3. **Extensive Experimental Validation:** We rigorously validated DiSRouter's generalizability and modularity. Our experiments extensively explored the self-assessment capabilities of both open-source and closed-source LLMs, quantified the superiority of intrinsic over external assessment, and verified the effectiveness of our training pipeline on both verifiable and open-ended tasks.
>
> ---
>
> For detailed experimental results and responses to reviewer-specific questions, please refer to our revised paper and our response to each reviewer. Thank you again for your time and effort.
>
> Sincerely,
>
> The Authors

---

### Meta-Review · Area_Chair_pt1r · 2025-12-14

**Summary:**

The following concerns remain outstanding and, in my view, prevent the work from being sufficiently solid:
1. Routing overhead, particularly time-to-first-token latency
2. The core utility metric does not account for cumulative latency

**Reviewer Concerns:**

**The following concerns have been addressed:**

1. Training efficiency and computational cost
2. Robustness evaluation
3. Clarification of training details
4. Performance with large-scale agent pools
5. Discussion of practical limitations
6. Presentation-related issues

**The following concerns have been mostly addressed:**

1. Generalization across different model types
2. Adaptation to open-ended tasks
3. Evidence supporting intrinsic assessment

**The following concerns remain outstanding:**

1. Routing overhead, particularly time-to-first-token latency
2. The core utility metric does not account for cumulative latency

**The following concern remains but is considered minor:**

1. Applicability to encoder–decoder models

**Reviewer Scores:**

1. Reviewer xhvr is expected to maintain the score as 6.
2. Reviewer qdad is expected to have a high chance of raising the score from 4 to 6.
3. Reviewer dfTT is expected to have a high chance of raising the score from 4 to 6.
4. Reviewer JRcd is expected to maintain the score as 8.

---

### Decision · Program_Chairs · 2026-01-26

Accept (Poster)